# Assimilating ESA-CCI Land Surface Temperature into the ORCHIDEE Land Surface Model: Insights from a multi-site study across Europe

Luis-Enrique Olivera-Guerra[1], Catherine Ottlé[1], Nina Raoult[2], Philippe Peylin[1]

[1]Laboratoire des Sciences du Climat et de l'Environnement, Institut Pierre-Simon Laplace (IPSL), CEA-CNRS-Université Paris-Saclay, Orme Des Merisiers, 91190 Gif-Sur-Yvette, France
[2]Department of Mathematics and Statistics, Faculty of Environment, Science and Economy, University of Exeter, Laver Building, North Park Road, Exeter, EX4 4QE, UK

*Correspondence to*: Luis Olivera-Guerra (luis.olivera@lsce.ipsl.fr)

**Abstract.** Land surface temperature (LST) plays an essential role in water and energy exchanges between Earth's surface and atmosphere. Recent advancements in high-quality satellite-derived LST data and land data assimilation systems present a unique opportunity to bridge the gap between global observational data and land surface models (LSMs) to better constrain the water/energy budgets in a changing climate. In this vein, this study focuses on the assimilation of the ESA-CCI LST product into the ORCHIDEE LSM (the continental part of the IPSL Earth System Model) with the aim of optimizing key parameters to improve the simulations of LST and surface energy fluxes. We use the land data assimilation system for the ORCHIDEE model (ORCHIDAS) to conduct a series of synthetic twin data assimilation experiments accounting for actual data availability and uncertainty from ESA-CCI LST to find an optimal strategy for assimilating LST. Here, we test different strategies of assimilation, notably investigating: i) two optimization methods (random-search and gradient-based) and ii) different ways to assimilate LST using the only raw data and/or different characteristics of the LST diurnal cycle (e.g., mean daily, daily amplitude, maximum and minimum temperatures, morning and afternoon gradients). Upon identifying the optimal approach, we use ORCHIDAS to assimilate ESA-CCI LST data across 34 European sites provided by the WarmWinter database. Our results demonstrate the effectiveness of assimilating 3-hourly CCI-LST data in ORCHIDEE over a single year in 2018, improving the accuracy of simulated LST and fluxes. This improvement, assessed against CCI-LST and in situ observations, reaches up to 60% reduction in root mean square deviation, with a median decrease of 20% over the entire validation period (2009-2020). Furthermore, we evaluate the effectiveness of optimized parameters for application at larger scales using the median of optimized parameters per vegetation type across sites. Notably, the performances for both LST and fluxes exhibit consistent stability over the years, comparable to using site-specific parameters, and indicate a significant improvement in the modeled fluxes. Future works will be focused on refining the utilization of the observation uncertainties provided by the ESA-CCI LST product (e.g., decomposed uncertainties and spatio-temporal variability) in the assimilation process.

## 1 Introduction

Surface heat fluxes, particularly sensible and latent heat fluxes, exchanged between the land surface and the atmosphere, play an essential role in the climate system as well as in numerous hydrological, meteorological, and agricultural applications (Bateni et al., 2013; Caparrini et al., 2004; Olioso et al., 1999). Land Surface Models (LSMs) have been developed to simulate the complex interactions between land surface and the atmosphere, offering valuable insights into the quantification and comprehension of energy and water fluxes within the system. While simulations from LSMs may accurately represent ecosystem state variables and observed fluxes under certain conditions, significant temporal and spatial biases still occur. These biases are attributed to simplifications in the representation of the processes governing energy and mass transfers, errors in the input data (atmospheric forcing, vegetation, and soil spatial information) and in the model parameterization.

In recent decades, efforts to address these discrepancies have explored the potential of using Land Surface Temperature (LST) observations through optimization or assimilation procedures in land surface monitoring (Boni et al., 2001; Coudert et al., 2008; Crow et al., 2003; Demarty et al., 2005; Ghent et al., 2010; Margulis and Entekhabi, 2003; Olioso et al., 1999; Ridler et al., 2012). In fact, LST is a key variable in LSMs because it reflects the coupled energy and water budgets, which are linked by evapotranspiration (Benavides Pinjosovsky et al., 2017; Coudert et al., 2008; Ridler et al., 2012). Therefore, LST observations have proven to be of great value not only for assessing water and energy fluxes but also in refining their estimation.

Recent advances in remote sensing technology and derived LST products provide a valuable benchmark for the evaluation and optimization of LSMs. Particularly noteworthy is the progress in high-quality satellite-derived LST that integrates measurements from diverse satellite sensors, including both geostationary and polar orbital platforms, such as the ESA CCI LST product (Hollmann et al., 2013; Veal et al., 2022). These advancements have yielded global LST products with enhanced spatial and temporal resolutions, thus expanding their potential utility and compatibility with LSMs.

A number of studies have been previously conducted to parameterize LSM using LST data (e.g. Boni et al., 2001; Coudert et al., 2008; Crow et al., 2003; Demarty et al., 2005; Ghent et al., 2010; Margulis and Entekhabi, 2003; Olioso et al., 1999; Ridler et al., 2012). Coudert et al. (2008) highlighted the advantages of incorporating the dynamics of the LST diurnal cycle for model parameter calibration compared to the direct use of raw LST observations. This is explained by the fact that the use of raw values of LST, prone to inaccuracies in their estimation, can introduce errors in the optimization process (Coudert et al., 2006; Demarty et al., 2005). This issue can be especially exacerbated when comparing different satellite products (Coudert et al., 2008). In this respect, the merged ESA-CCI LST product derived from various satellite thermal infrared sensors allows to assess the diurnal cycle characteristics given its harmonization between LST satellite products, while providing a global product with associated uncertainties. The availability of an LST global product with its corresponding uncertainties is an important asset for the data assimilation (DA) processes in LSMs since the knowledge of observations and model uncertainties are crucial in DA to obtain realistic model-data fits.

DA offers a valuable framework for integrating measurements and models, weighting the sources of error in both, to generate a statistically optimal and dynamically consistent estimation of the evolving system state (Margulis and Entekhabi, 2003). Some studies have been focused on the assimilation of LST into LSMs aiming to enhance simulations of LST and water and energy fluxes (Bateni et al., 2013; Benavides Pinjosovsky et al., 2017; Caparrini et al., 2003; Ghent et al., 2010; Lu et al., 2017; Meng et al., 2009; Sini et al., 2008). DA approaches can also be used to estimate model parameters (Rayner et al., 2019), either independently, or together with the model state (Bateni and Entekhabi, 2012; Moradkhani et al., 2005). Therefore, DA techniques have been widely used to control LSMs state variables and/or for parameters optimization ( e.g. Bacour et al., 2023; Moradkhani et al., 2005; Peng et al., 2011; Raoult et al., 2016; Rayner et al., 2005; Santaren et al., 2007). DA techniques allow us to calibrate LSMs against observational data, providing best-fit internal parameters and associated uncertainty ranges compared to default parameters.

Recent advancements in both high-quality satellite-derived LST data and land DA systems offer a promising opportunity to bridge the gap between observations and LSMs, thereby to better constrain the land surface water and energy budgets. In this vein, this study focuses on assimilating the merged LST data from the ESA-CCI product into the ORganizing Carbon and Hydrology In Dynamic EcosystEms (ORCHIDEE) LSM, the continental component of the IPSL (Institut Pierre Simon Laplace) Earth System Model. Concurrently, there is a growing availability of in situ measurement sites providing estimates of sensible and latent heat fluxes at high temporal resolution, primarily from FluxNet observations (FLUXNET, 2016). Recognizing the value of both data sources, this study aims to leverage the complementary information from CCI-LST product and WarmWinter database (Warm Winter 2020 Team and ICOS Ecosystem Thematic Centre, 2022). We achieve this by assimilating CCI-LST data at selected sites from the WarmWinter database and utilizing fluxes observations as independent

validation data. Our primary objective is to investigate whether LST observations from the ESA-CCI product have the potential to improve ORCHIDEE simulations of water and energy fluxes, accounting for their frequency and measurement uncertainties. To this end, we seek to identify an optimal assimilation strategy by testing different approaches, including assimilating raw LST observations and specific characteristics of the LST diurnal cycle (e.g. daily maximum, amplitude, morning and afternoon gradients). We investigate whether assimilating observed characteristics of the LST diurnal cycle can provide additional constraints compared to assimilating raw LST data alone. We initially conducted a series of synthetic twin DA experiments to select the optimal strategy of assimilation. Then, we implemented the selected DA strategy across 34 European sites provided by the WarmWinter database using ESA-CCI LST data extracted at each site, assessing the effectiveness of the assimilation process in improving LST and surface energy flux simulations.

## 2 Materials and methods

### 2.1 Data

#### 2.1.1 LST observations from the ESA-CCI product

We use in this study, the merged Infrared (IR) Climate Data Records (CDR) from the LST Climate Change Initiative (CCI) project by the European Space Agency (ESA) (Hollmann et al., 2013; Veal et al., 2022), referred to as CCI-LST. The merged IR CDR includes all available IR geostationary sensor data and IR Low-Earth Orbiting (LEO) sensor data over the period 2009-2020 being delivered with a 3-hourly temporal resolution and global spatial resolution of 0.05°. The IR geostationary sensor includes the Imager on the Geostationary Operational Environmental Satellite (GOES) platforms, the Spinning Enhanced Visible and Infrared Imager (SEVIRI) onboard the Meteosat Second Generation platforms and the Japanese Advanced Meteorological Imager (JAMI) on the Multi-functional Transport Satellite (MTSAT) platform. The LEO data come from Advanced Along-Track Scanning Radiometer (AATSR), the MODerate resolution Imaging Spectroradiometer (MODIS) onboard Terra and Aqua, and the Sea and Land Surface Temperature Radiometer (SLSTR) on board Sentinel-3A and -3B.

The CCI-LST observations have an associated total uncertainty estimate derived from different error components. These error components correlate on various spatial and temporal scales, such as: i) random uncertainties weakly correlated (like random noise in the satellite data), ii) locally correlated atmospheric uncertainties (related to atmospheric conditions), iii) locally correlated biome or surface uncertainties, iv) large scale systematic uncertainties (related to calibration of the satellite sensor) and v) locally correlated LST correction uncertainties (for inter-calibration or time corrections). The total uncertainty is obtained from the sum of each uncertainty component in quadrature (i.e., the square root of the sum of squares), which is used in this study to prescribe observation errors in the assimilation process (Sect. 2.3).

Figure 1 illustrates the median of 3-hourly CCI-LST uncertainties during the optimization year (2018) at the corresponding CCI-LST pixel of 0.05° over each of the 34 selected sites, detailed below. These uncertainties are used in the assimilation process as the observation error. The median LST uncertainties are, on average, 1.05 K across the 34 sites evaluated, ranging from 0.76 K to 1.89 K for DE-Kli and IT-BCi sites, respectively. It should be noted that the Mediterranean sites present larger uncertainties ranging between 1.50 K and 1.89 K. These larger values are mainly explained by the LST correction component (i.e., the (v) uncertainty component mentioned above), which could be attributed to more complex surface conditions, such as heterogeneous land cover types.

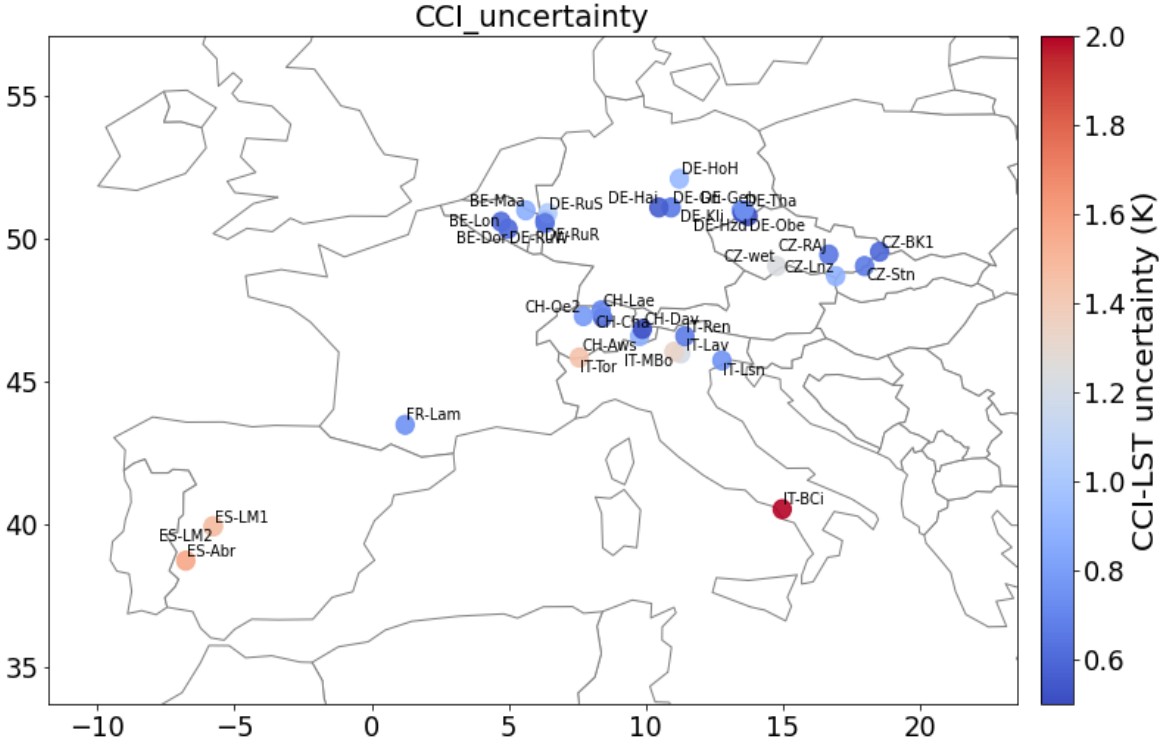

**Figure 1. Location of the eddy-covariance flux tower sites from the WarmWinter network. The median of the 3-hourly LST uncertainty from the ESA-CCI product during the optimization year (2018) is represented at the corresponding CCI-LST pixel.**

## 2.1.2 In situ observations

We use meteorological data and measurements of surface energy fluxes of the stations available in Europe from the WarmWinter database (Warm Winter 2020 Team and ICOS Ecosystem Thematic Centre, 2022). The sites used in this study are depicted in Fig. 1 and detailed in Table A1 (location, data period, vegetation and climate types). The WarmWinter network
is chosen for its coverage of recent periods, which aligns with the CCI-LST period, and it represents the latest freely available dataset for the EU. The meteorological data (air temperature, humidity, pressure, wind speed, rainfall and snowfall rates, shortwave and longwave incoming radiation) at a 30-minute time step is used as forcing for the ORCHIDEE model. The eddy-covariance measurements of latent heat (LE) and sensible heat (H) fluxes, as well as net radiation (Rn), are also used to evaluate the performance of ORCHIDEE simulations using default parameters and after optimization. We selected these 34 sites based
on two criteria: i) the dominant vegetation type accounting for more than 50% of the corresponding ORCHIDEE grid cell, and ii) full data availability for the optimization year (2018) across all sites. We chose the year 2018 for two main reasons: i) all sites have recorded observations during this year, and ii) there are events of drought in Europe throughout certain periods. This selection enables the consideration of parameters associated with droughts in both the sensitivity analysis and potentially in the optimization process. The selected sites collectively represent 7 out of the 15 Plant Functional Types (PFTs) of ORCHIDEE
and the length of each observation record varies from 4 to 11 years (considering only the available period of CCI-LST data). Thus, the selection of 34 sites ensures a comprehensive representation of diverse land vegetation types, being as homogeneous as possible at the 0.05° resolution of the CCI-LST product.

## 2.2 The ORCHIDEE land surface model

The ORCHIDEE land surface model is designed to simulate the exchanges of carbon, water, and energy between the land surface and the atmosphere, as detailed by Krinner et al. (2005). In this study, we use version 2.2, which has been developed at the IPSL (Institut Pierre Simon Laplace, France) and contributed to the Climate Model Intercomparison Project 6 (CMIP6) under a coupled configuration with an atmospheric circulation model (Boucher et al., 2020). The ORCHIDEE model allows an implementation across a broad range of spatial scales - including grid-point, regional or global levels - and spans timescales ranging from 30 minutes to thousands of years.

The study employs a temporal resolution of half-hourly intervals to model hydrological and photosynthetic processes, as well as the energy balance. In contrast, slower components related to carbon allocation within plants, autotrophic respiration, leaf onset and senescence, plant mortality, and soil organic matter decomposition are treated at a daily time step. The hydrological model in ORCHIDEE discretizes the first 2 m of the soil column into 11 layers, each with increasing grid spacing progressing geometrically with a ratio of 2 (de Rosnay et al. 2002). The soil moisture at various levels is determined by solving the Richards equation, which models the vertical water transfers in the unsaturated zone. The land surface in ORCHIDEE is characterized by 15 PFTs, including bare soil, which can coexist within a given grid cell. The yearly varying PFT maps are derived from the ESA CCI land cover products (Lurton et al., 2020; Poulter et al., 2015). In this study, the soil types are characterized by soil textures for which ORCHIDEE uses the global soil map based on the Zobler soil classification (Zobler, 1986) reduced to 3 different textures. Except for phenology, the processes are described by generic equations for the different vegetation and soil types but with different parameters that are PFT-specific or soil texture-specific, respectively.

We implement ORCHIDEE at site level using half-hourly meteorological data measured at each of the 34 sites evaluated. A prior spin-up simulation was performed at each site to bring soil carbon pools, vegetation state and soil moisture content to equilibrium. This procedure is applied by running ORCHIDEE for several hundred years recycling the available forcing data with present-day $CO_2$ concentration. A transient simulation (starting from the first year of measurement for each data stream) was then performed after each spin-up simulation, accounting for the secular increase of atmospheric $CO_2$ concentrations.

## 2.3 Parameter optimization methodology

### 2.3.1 Data Assimilation framework

The ORCHIDEE Data Assimilation System (ORCHIDAS) has been extensively discussed in previous studies (Bacour et al., 2015; Bastrikov et al., 2018; Kuppel et al., 2012; Peylin et al., 2016; Santaren et al., 2014). This assimilation framework is built upon a variational Bayesian approach, optimizing ORCHIDEE parameters represented by a vector $x$. The optimization process involves iteratively minimizing a global cost function $J(x)$ (Tarantola, 2005), assuming Gaussian errors for both observations and model parameters:

$$J(x) = \frac{1}{2}[(H(x) - y)^T R^{-1}(H(x) - y) + (x - x_b)^T B^{-1}(x - x_b)] \tag{1}$$

where $x$ is the vector of parameters to optimize and $y$ is the vector of observations. The first part of the cost function measures the mismatch between the observations ($y$) and the corresponding model outputs ($H(x)$), and the second part represents the mismatch between the prior parameter values ($x_b$) and the optimized parameters x. T represents the transpose of the matrix. Both terms of the cost function are weighted by their prior covariance matrices: $R$ and $B$ for the observation and parameter errors, respectively. Since the error covariance matrices are difficult to assess, they are neglected in this study, so $R$ and $B$ are

diagonal as in most studies. The observation errors for each site are prescribed as the median of CCI-LST uncertainty over the optimization year (2018) for the corresponding pixel, and are illustrated in Fig. 1.

Two methods to minimize the cost function are tested in this study: gradient-based and random search algorithms, described below.

### Gradient-based minimization algorithm: L-BFGS-B

We use the gradient-based L-BFGS-B (limited memory Broyden-Fletcher-Goldfarb-Shanno algorithm with bound constraints; Byrd et al., 1995), referred to as BFGS, to iteratively minimize $J(x)$. BFGS offers the advantage of considering bounds in
parameter variations, enabling comparability with the genetic algorithm and leveraging our existing knowledge of parameters. The algorithm requires the evaluation of $J(x)$ and its gradient with respect to each parameter to explore the parameter space. In this study, the gradient is calculated with a finite-difference approximation, where we quantify the ratio between the alteration in model output and the adjustment in model parameter. The search ceases once the relative change in $J(x)$ falls below $10^{-4}$ for five consecutive iterations. Since gradient-based algorithms have the potential drawback of converging to local
minima rather than to the global one (more likely in non-linear models), we run a set of independent assimilations initiated with 16 distinct random initial guesses for the parameter vector, as in Bastrikov et al., (2018). Although some issues may arise when using Gaussian assumptions in gradient-based minimization algorithms (MacBean et al., 2016), most parameter errors follow Gaussian distributions in the cases of ORCHIDEE (Santaren et al., 2007).

### Random search minimization algorithm: Genetic algorithm

Genetic algorithm (GA) method is derived from the principles of genetics and natural selection (Goldberg, 1989; Haupt and Haupt, 2004) that performs a stochastic search over the entire parameter space. Parameter vectors are considered as chromosomes, with each gene corresponding to a given parameter. The algorithm works iteratively, filling a pool of a given number of chromosomes at each iteration. The initial pool is created with randomly perturbed parameters. Then, a sequence of operations (selection, crossover, and mutation) simulates population evolution, which is described for ORCHIDAS in
Santaren et al., (2014) and Bastrikov et al., (2018). We adopt the identical GA configuration as Santaren et al., (2014) who experimented various GA setups to identify the setup yielding the smallest optimal cost function and requiring the fewest iterations. The configuration consists of a population of 30 chromosomes, a maximal number of iterations of 40, a crossover/mutation ratio of 4:1, a number of gene blocks exchanged during crossover of 2 and a number of genes perturbed during mutation of 1.

**2.3.2 Sensitivity analysis and parameters to be optimized**

We perform sensitivity analysis at each of the 34 sites assessed to determine which ORCHIDEE parameters have the most influence on the simulated LST and characteristics of the diurnal cycle. The selection of crucial parameters for optimization serves a dual purpose: it enhances computational efficiency in the optimization process and mitigates the risk of overfitting.

We use the Morris method (Morris, 1991) to assess the sensitivity of 19 parameters linked to ORCHIDEE LST simulations,
detailed in Table 1. We implement this method since it is effective with relatively few model runs compared to other methods like the Sobol' sensitivity analysis (Sobol, 2001). Furthermore, the Morris method has been frequently utilized for parameter selection in ORCHIDEE (e.g. Abadie et al., 2022; Bastrikov et al., 2018; Raoult et al., 2021, 2023). The Morris method is a one-factor-at-a-time (OAT) method of sensitivity analysis that evaluates the relative importance of the parameters from elementary effects (EE) of each parameter on model outputs. Basic statistics from multiple EEs in the parameter space allow
us to build a ranking, which approximates well to a global sensitivity measure. This qualitative method requires only a small number of simulations, $(p+1) \times n$, where p is the number of parameters and $n$ is the number of random trajectories generated.

Since the sampling strategy significantly impacts the Morris method, we apply the improved sampling strategy proposed by Campolongo et al., (2007) suited for models with a large number of parameters like ORCHIDEE. This strategy maximizes the dispersion of trajectories in the parameter space. In addition, the number of trajectories and levels (i.e. the sampling of the parameter space) can impact considerably the method and the selection of parameters to be optimized (Ruano et al., 2012). Therefore, we conducted a preliminary test to select the optimal $n$ trajectories for the ORCHIDEE model by evaluating the Morris results by increasing $n$ from 5 to 100 according to the position factor of the rankings proposed by Ruano et al., (2012). Based on this test, we use 20 trajectories and 10 levels for an optimal sampling of the parameter space. The Morris method applied to each site allows to identify the most sensitive parameters for each site, which are subsequently retained for the single-site optimization in the DA experiments, described in Sect. 2.4. An example of these outcomes is presented in Fig. B1 (see Appendix) over a selected site in Spain (ES-Abr) for the twin DA experiments, whose identified parameters to be optimized are highlighted in bold in Table 1. The sensitivity analysis at the ES-Abr site required 400 simulations, (19+1)x20.

The mean ($\mu$) and standard deviation ($\sigma$) for all the trajectories are calculated to assess the overall importance of each parameter. We select the parameters with a normalized $\mu$ or normalized $\sigma$ value higher than 0.2 in the twin experiment and at the rest of the sites. For the 34 sites, the same 19 parameters are evaluated in the sensitivity analysis, from which the number of selected parameters to be optimized ranges between 5 and 17, with one-third of the sites presenting 12 parameters to be optimized.

**Table 1.** ORCHIDEE parameters evaluated in the sensitivity analysis at the 34 sites: model parameter name, descriptions and default values are shown. The 11 sensitive parameters to be optimized over the selected site (ES-Abr) for the twin DA experiments are indicated in bold. For the twin experiments, absolute values for soil hydrology parameters are used, whereas scaling factors are applied to the 34 sites (values in parentheses) since they depend on soil type.

| Parameter | Description | Prior | Minimum | Maximum |
|---|---|---|---|---|
| **Energy balance** | | | | |
| **Albedo\*** | Scaling factor for surface albedo | 1 | 0.75 | 1.25 |
| **STC\*** | Scaling factor for soil thermal conductivity | 1 | 0.7 | 1.3 |
| **HC\*** | Scaling factor for soil heat capacity | 1 | 0.7 | 1.3 |
| **$r_{soil}$\*** | Scaling factor for soil resistance to evaporation | 1 | 0.5 | 1.5 |
| **z0** | Bare soil roughness length | 0.01 | 0.001 | 0.015 |
| **Soil Hydrology** | | | | |
| **$\theta_{res}$\*** | Scaling factor for residual soil moisture | 0.078 (1) | 0.062 (0.7) | 0.083 (1.3) |
| **$\theta_{sat}$\*** | Scaling factor for soil moisture at saturation | 0.43 (1) | 0.34 (0.8) | 0.52 (1.2) |
| A\* | Scaling factor for van Genuchten coeff a | 0.0036 (1) | 0.0022 (0.6) | 0.0050 (1.4) |
| **n\*** | Scaling factor for van Genuchten coeff n | 1.56 (1) | 1.10 (0.7) | 2.20 (1.3) |
| $Sl_r$ | Slope coefficient for re-infiltration | 0.5 | 0.1 | 2 |
| **Soil water availability** | | | | |
| $\theta_{crit,rel}$ | Relative soil moisture above which transpiration is maximal | 0.8 | 0.3 | 0.9 |
| **z** | Root profile | 4 | 1 | 10 |

| | | | | |
|---|---|---|---|---|
| $\alpha$ | Controls water stress curve | 1 | 0.05 | 10 |
| **Photoshynthesis** | | | | |
| B1 | Factor for calculation of leaf-to-air vapor pressure difference | 0.22 | 0.11 | 0.44 |
| $VC_{max25}$ | Maximal rate of Rubisco activity-limited carboxylation at 25° | 50 | 30 | 80 |
| **Phenology** | | | | |
| $h_{veget}$ | Prescribed vegetation height | 1 | 0.5 | 1.5 |
| $LAI_{max}$ | Maximum leaf area index | 5 | 3 | 8 |
| **$L_{age,crit}$** | Critical leaf age (days) | 120 | 30 | 150 |
| **SLA** | Specific leaf area ($m^2g^{-1}$) | 26 | 13 | 0.05 |

* Scaling factor applied to an ORCHIDEE parameterization. Note that the scaling factor may involve more than a unique parameter in the model.

### 2.3.3 Selection of the DA experiment set up

A series of synthetic twin DA experiments are tested over a selected site in Spain (ES-Abr) to understand the respective constraint brought by LST pseudo-observations as well as several characteristics of the diurnal cycle of LST and evaluate their capability to improve ORCHIDEE simulations. The characteristics of the diurnal cycle of LST are estimated based on 3-h interval model outputs to be consistent with the 3-h frequency of CCI-LST data. The characteristics evaluated are daily LST amplitude, maximum LST, minimum LST, morning gradient (slope between 10:00 and 13:00 LT (local time)) and afternoon

gradient (slope between 16:00 and 19:00 LT). Additionally, we evaluate assimilating the LST at 13:00 LT since the early afternoon is well-suited for detecting water stress (Lagouarde and Bhattacharya, 2018; Koetz et al., 2018). This is particularly relevant because the upcoming TRISHNA mission will provide LST at 13:00 LT, a few hours later than other thermal missions widely used in recent decades for stress detection (e.g., Landsat, ASTER, MODIS). Consequently, we aim to assess the potential impact of assimilating a single LST observation per day at this particular time. LST pseudo-observations and each

derived characteristic of the diurnal cycle are assimilated separately, and then different combinations are considered, some of which are detailed in Table 2. All the combinations tested are presented in Table S.1 (see in supplementary material).

In each DA experiment, we optimize the 11 parameters selected from the sensitivity analysis carried out over the ES-Abr site for 3-hourly LST and each characteristic (parameters in bold in Table 1). The LST pseudo-observations for the DA experiments are generated by running the ORCHIDEE model using its default parameter values. A set of random values for the 11

parameters to be optimized is then used as the prior parameters in the optimizations. No uncertainties or model discrepancies are considered in these first LST pseudo-observations (unbiased by observational uncertainties) to allow us to directly assess the performance and convergence properties of the optimization schemes. In these DA experiments, the second term of the cost function (Eq. 1) was excluded to evaluate the role of observations. Therefore, we can conduct a direct comparison between the optimized parameters and the 'true' parameters − set as the ORCHIDEE parameters by default − enabling the optimization

process to freely recover these parameters and assess the accuracy of the optimizations.

We perform 16 independent runs for each DA experiment with 16 different first guesses randomly selected within the range of variation of the parameters, as Bastrikov et al., (2018). The choice of 16 sets is a trade-off between computational cost and minimizing the risk of not converging toward a stable cost function minimum, as verified by Bastrikov et al., (2018). The data assimilation experiments are tested using the BFGS and GA methods starting from the 16 sets of different first guesses. The

16 different first guesses are kept identical for all the DA experiments (including between BFGS and GA methods) to ensure the same first-guess values and to be consistent between DA experiments.

The prior uncertainty on the ORCHIDEE parameters is set to 15% of the range of variation for each parameter. This error is set following preliminary DA tests over the ES-Abr site, where we explored different percentages of the range of variation as parameter errors. We found that a prior parameter error ranging between 10% and 20% of the range of variation emerges as optimal, exhibiting further improvements in fluxes after assimilating LST. Unlike the assimilation of actual CCI-LST data where we used the errors provided with the LST product (see Sect. 2.3.4), in the twin DA experiments the observation errors are defined as the mean-squared difference between the observations and the prior model simulations, following Bastrikov et al., (2018).

The performances of the DA experiments are assessed in terms of their ability to: i) retrieve 'true' parameter values and, ii) simulate the half-hourly LST and turbulent fluxes from the posterior parameters. The performances are evaluated using the reduction in the root mean square difference (RMSD) between the prior and the posterior, as defined in Eq. (2). In addition, the performance is evaluated by comparing the posterior parameter values and against the true values, as estimated through the pseudo-observation tests.

$$RMSD_{reduction} = \left(1 - \frac{RMSD_{post}}{RMSD_{prior}}\right)100 \qquad (2)$$

For each DA experiment, independent of the observational constraint assimilated, the $RMSD_{reduction}$ is estimated for the half-hourly LST and turbulent fluxes (LE and H) simulations. This evaluation involves calculating the model-data fit between each posterior simulation using 16 different first-guesses and the prior simulations (from ORCHIDEE run with a unique set of prior parameters).

Based on the outcomes of the DA pseudo-data experiments, we select those showing the best results fitting the LST and turbulent fluxes. To evaluate the feasibility of assimilating the CCI-LST product, we consider its actual availability and uncertainties within the chosen optimal strategies. Thus, we implement a second set of twin DA experiments introducing some modifications. The LST pseudo-observations are filtered out with the actual availability of CCI-LST, thereby accounting for gaps in the time series resulting, for example, from cloudy conditions. Subsequently, these pseudo-observations are perturbed with Gaussian errors derived from their respective 3-hourly CCI-LST uncertainties. This refined approach ensures a more realistic representation of the assimilation process, considering both the real-world constraints and uncertainties associated with the CCI-LST data.

**Table 2.** Example of typical ORCHIDEE variables optimized in the DA experiments performed and used to determine the optimum strategy.

| Experiment name | Selected observational constraint |
| --- | --- |
| LST13 | LST at 13:00 local time (12:00 UTC) |
| Tmax | Maximum daily LST over 3-hourly interval |
| Ampl | Daily amplitude LST over 3-hourly interval |
| slope13 | Temporal gradient between 10:00 and 13:00 local time |
| 3h-LST | 3-hourly LST series |

| 3h-LST+Tmax | LST + Tmax |
|---|---|

**2.3.4 Assimilation experiments based on ESA CCI-LST data**

Once the optimal DA strategy is selected (from the twin experiments at one site), it is implemented across the 34 European
sites of the WarmWinter2020 network for the year 2018. The parameters to be optimized are identified per site from the
sensitivity analysis described in Sect. 2.3.2. The performances are evaluated mainly from RMSD reduction between the prior
and the posterior ORCHIDEE simulations against CCI-LST and in situ energy fluxes (Rn, LE and H) observations, described
in Sect. 2.1. Additionally, we employ the decomposition of the Mean Square Error (MSE: Kobayashi and Salam, 2000) to gain
deeper insights into which specific error components are improving or degrading. The MSE (i.e. the square of the RMSD) is
decomposed into three components: the squared bias (SB), the lack of correlation weighted by the standard deviation (LCS)
and the squared difference between standard deviations (SDSD). The three components are expressed as:

$$SB = (\overline{y_{sim}} - \overline{y_{obs}})^2 \tag{3}$$

$$LCS = 2\sqrt{\frac{1}{N}\sum_{i=1}^{N}(y_{sim,i} - \overline{y_{sim}})^2}\sqrt{\frac{1}{N}\sum_{i=1}^{N}(y_{obs,i} - \overline{y_{obs}})^2}(1 - r) \tag{4}$$

$$SDSD = \left(\sqrt{\frac{1}{N}\sum_{i=1}^{N}(y_{sim,i} - \overline{y_{sim}})^2} - \sqrt{\frac{1}{N}\sum_{i=1}^{N}(y_{obs,i} - \overline{y_{obs}})^2}\right)^2 \tag{5}$$

with $\overline{y_{sim}}$ and $\overline{y_{obs}}$ being the means of simulations ($y_{sim,i}$) and observations ($y_{obs,i}$), respectively, and $r$ the correlation coefficient.
The MSE decomposition provides valuable information on areas of improvement and potential limitations in the assimilation
process.

The parameters optimized in 2018 are then used to evaluate ORCHIDEE simulations during the validation period from 2009
to 2020 (i.e. the CCI-LST period), where in situ observations are available. Additionally, we evaluate a unique set of parameters
per PFT from the previous site-specific-optimized parameters in 2018. This is a key step in scaling parameter optimization
since broader-scale simulations (i.e. from regional up to global) require a unique set of parameters per PFT. Thus, we calculate
the median of optimized parameters across sites to evaluate the effectiveness of optimized parameters at larger scales. The
median of generic parameters is computed across all sites, while PFT-specific parameters are computed per PFT. Finally, we
run ORCHIDEE with this unique parameter set per PFT over the 34 sites from 2009 to 2020. This assessment aims to determine
whether the performances at each site align with those achieved using site-specific optimized parameters. Note that we did not
use the multi-site optimization approach as in Kuppel et al., (2012), given that several parameters concerning the soil thermal
and hydrological properties would complicate the setup of such a configuration (sites of a given PFT have different soil
properties).

## 3 Results

### 3.1 Using pseudo-data twin experiments to define the optimization strategy

#### 3.1.1 Model−data fit and fluxes improvement

To investigate the differences between the assimilated observational constraints (as detailed in Table 2), Fig. 2 compares the overall optimization performances for the DA experiments. It displays the distribution of the RMSD reduction in half-hourly LST, LE and H between the prior (ORCHIDEE run with a unique set of parameters) and posterior model after assimilation, with each DA experiment including 16 first-guess tests conducted at the selected ES-Abr site in Spain. The model improvement

is represented by the spread of boxplots for different initial first guesses, with each box corresponding to the assimilation of different characteristics of the diurnal cycle and their combinations. As expected, assimilating LST leads to improvements of LE and H fluxes, affirming the significance of LST data in constraining the turbulent fluxes. Furthermore, it is highlighted that the improvement in H is larger than that of LE in each DA experiment due to its more direct dependence on LST.

Overall, it becomes evident that the BFGS method is more reliant on initial first guesses, shown by the larger spread in each

DA experiment for the three analyzed variables (LST, LE, and H). This confirms the higher likelihood of gradient-based methods to get stuck in local minima. Notably, the GA method consistently outperforms the BFGS method, yielding more substantial improvements with RMSD reduction surpassing that achieved by BFGS in every experiment and for each variable. Remarkably, the GA method shows improvements across all three variables and each of the 16 runs achieves RMSD reductions greater than 0. Conversely, certain optimizations employing the BFGS method result in deteriorated simulations after the

assimilation, with negative RMSD reduction for the half-hourly LST and LE. This is obtained mainly when the amplitude (Ampl) and morning slope (slope13) of the LST diurnal cycle are assimilated. This underscores the superior reliability and performance consistency of the GA method compared to the BFGS method across the evaluated variables and experiments.

Concerning improvements in half-hourly LST simulations through various DA experiments, superior performances are observed when assimilating the full LST series alone or incorporating a characteristic of the diurnal cycle in the assimilation

(utilizing Tmin, Tmax and/or Ampl, see supplementary Fig. S1). Nevertheless, incorporating Tmin, Tmax and/or Amplitude to the full LST series results in only marginal additional improvement in half-hourly LST simulations. In fact, the median reductions in RMSD remain consistently similar, around 76-79%, across all experiments using the full LST series as a constraint. Contrastingly, the assimilation of a single LST observation per day - such as LST at 13:00 h (LST13) or maximum daily LST (Tmax) only - leads to significantly less improvement in half-hourly LST simulation. Median reductions in RMSD

are 26% for LST13 and 45% for Tmax, accompanied by notably larger spreads, similar to those obtained with the BFGS method.

Regarding the improvements in half-hourly simulations of the surface turbulent fluxes (LE and H), the results are much more stable than those obtained for LST across the different DA experiments. This stability is observed in terms of median reductions in RMSD as in distributions across the 16 independent runs. While the median reductions in RMSD of LE range between 55%

and 72% for slope13 and 3h-LST+Tmax experiments, respectively, the improvements in H range between 71% and 87% for LST13 and 3h-LST+Tmax, respectively. It should be noted that assimilating jointly Tmax and LST (3h-LST+Tmax) improves considerably the LE and H simulations compared to assimilating LST only, with 59% and 81% RMSD reduction for LE and H, respectively. The 3h-LST+Tmax not only offers superior performance in simulating H but also produces the least dispersion in RMSD reduction. As can be seen, larger improvements using the GA method are obtained in H simulations, with higher

RMSD reduction and smaller spreads than those depicted for LST and LE, as can be noticed from the interquartile ranges.

Summarizing the results focusing only on the GA results and the three analyzed variables, the most substantial enhancements are evidenced when considering the entire 3-hourly LST series, either independently (3h-LST DA experiment) or jointly with

other attributes of the diurnal cycle such as the 3h-LST+Tmax DA (or 3h-LST+Ampl and 3h-LST+Ampl+Tmax experiments not shown). These configurations yield an average RMSD reduction ranging between 72% and 78% for the three variables

(LST, LE and H). Conversely, assimilating a single characteristic of the diurnal cycle (LST13, Tmin, Tmax, Ampl, slope13 and slope19 DA experiments) results in comparatively smaller improvements, with an average RMSD reduction ranging between 53% and 67%. Furthermore, it is worth noting that assimilating a single daytime observation independently of all other observations throughout the day, specifically to calculate diurnal cycle characteristics (e.g., LST at 13:00), yields the least improvement in LST posterior simulations. While considering the potential assimilation of data from upcoming thermal

missions like TRISHNA or LSTM, which will provide LST at 13:00 primarily to detect water stress (Lagouarde and Bhattacharya, 2018; Koetz et al., 2018), our findings suggest that assimilating LST13 in ORCHIDEE is less optimal to enhance the diurnal cycle of LST and energy fluxes, compared to other constraints. It is interesting to compare the performances obtained by assimilating a single LST at 13:00 (LST13) with the daily maximum LST (Tmax), which typically occurs at 16:00 on a 3-hourly basis at the ES-Abr site. The LST13 experiment results in lower improvements in LST and H simulations than

when assimilating Tmax. Specifically, the median RMSD reductions in LST simulations are 26% for the Tmax experiment and 45% for the LST13 experiment. Conversely, both experiments (Tmax and LST13) yield nearly identical median RMSD reductions (63% and 64%) in LE simulations, slightly higher than assimilating the entire 3-hourly LST time series (59%). In any case, the fact that the errors in the surface fluxes and LST simulations have been reduced by assimilating a single observation at 13:00 confirms the potential use of TRISHNA or LSTM for monitoring water resources.

Nevertheless, through the combination of different characteristics of the diurnal cycle, such as in the LST13+slope13+slope19 and Ampl+LST13+slope13+slope19 DA experiments (not shown), noteworthy improvements similar to those achieved using the entire LST series can be attained, with an average RMSD reduction of 74% for both experiments.

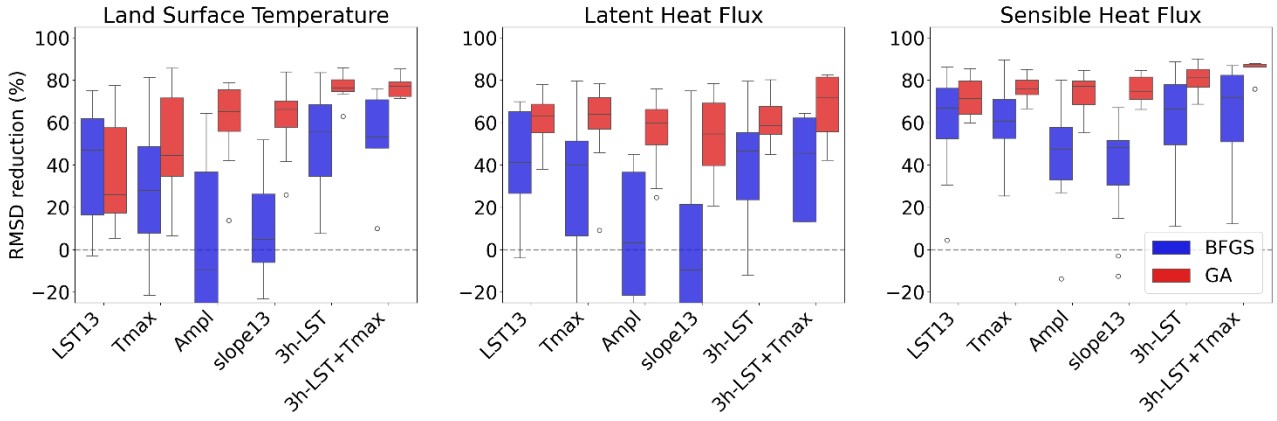

**Figure 2. Boxplots obtained within 16 optimization tests with random first-guess parameter values for each DA experiment over the selected site in Spain (ES-Abr) comparing the performances between the gradient-based (in blue) and genetic (in red) methods in terms of model–data RMSD reduction (%) obtained for 30-min LST, LE and H. The x-axis indicates the DA experiment assimilating a single characteristic of the diurnal cycle (LST13, Tmax, Ampl and slope13), the 3-hourly LST pseudo-observations (LST) and including the Tmax (LST+Tmax). The RMSD reduction (%) is computed between the 16 posterior simulations per DA experiment and the prior simulations (generated from a single set of prior parameters).**

### 3.1.2 Parameter and uncertainty estimates

The optimized parameters from the different DA experiments are illustrated in Fig. 3. In general, when utilizing the GA
method, the averages of the optimized parameters from the 16 independent runs per DA experiment closely align with the 'true'
parameter values for a majority of the parameters. This holds especially for n, $L_{age,crit}$ and z, which are among the most LST-
sensitive parameters according to the sensitivity analysis (see Fig. B1 in Appendix). Furthermore, irrespective of the DA
experiment, these three parameters exhibit the smallest standard deviation. This indicates that the GA method consistently
converges accurately to the 'true' parameter values across the 16 runs. It is worth noting that the most sensitive parameter
(Albedo*) does not show stable results across DA experiments using the GA method. While Albedo* shows mean values in
close proximity to the 'true' value, it exhibits larger spreads when assimilating single characteristics of the diurnal cycle, such
as those observed with the BFGS method. This behavior can be attributed to the nature of this surface albedo parameter, which
was defined as a multiplicative factor applied to two albedo components associated with vegetation and soil. This configuration
may lead to error compensation effects between several parameters when assimilating a single characteristic, preventing a
unique solution for Albedo* in the optimization process. Nevertheless, when assimilating a combination of four or five
characteristics of the diurnal cycle, such as Ampl+LST13+slope13+slope19 and Ampl+Tmax+LST13+slope13+slope19 DA
experiments (see supplementary Fig. S2), the obtained results are comparable to assimilating the 3-hourly LST alone. In these
cases, the mean optimized values are very close to the true value, and the spreads are smaller, indicating a more accurate and
consistent assimilation outcome.

Even though HC* (related to soil heat capacity) and STC* (related to soil thermal conductivity) are among the most sensitive
parameters to the selected characteristics of the diurnal cycle (Fig. B1), their retrieval is not optimal for all DA experiments,
as indicated by large spreads in their optimized values. This is directly linked to the fact that they are highly anticorrelated
with other parameters, as evidenced by the posterior covariance matrix (see supplementary Fig. S3).

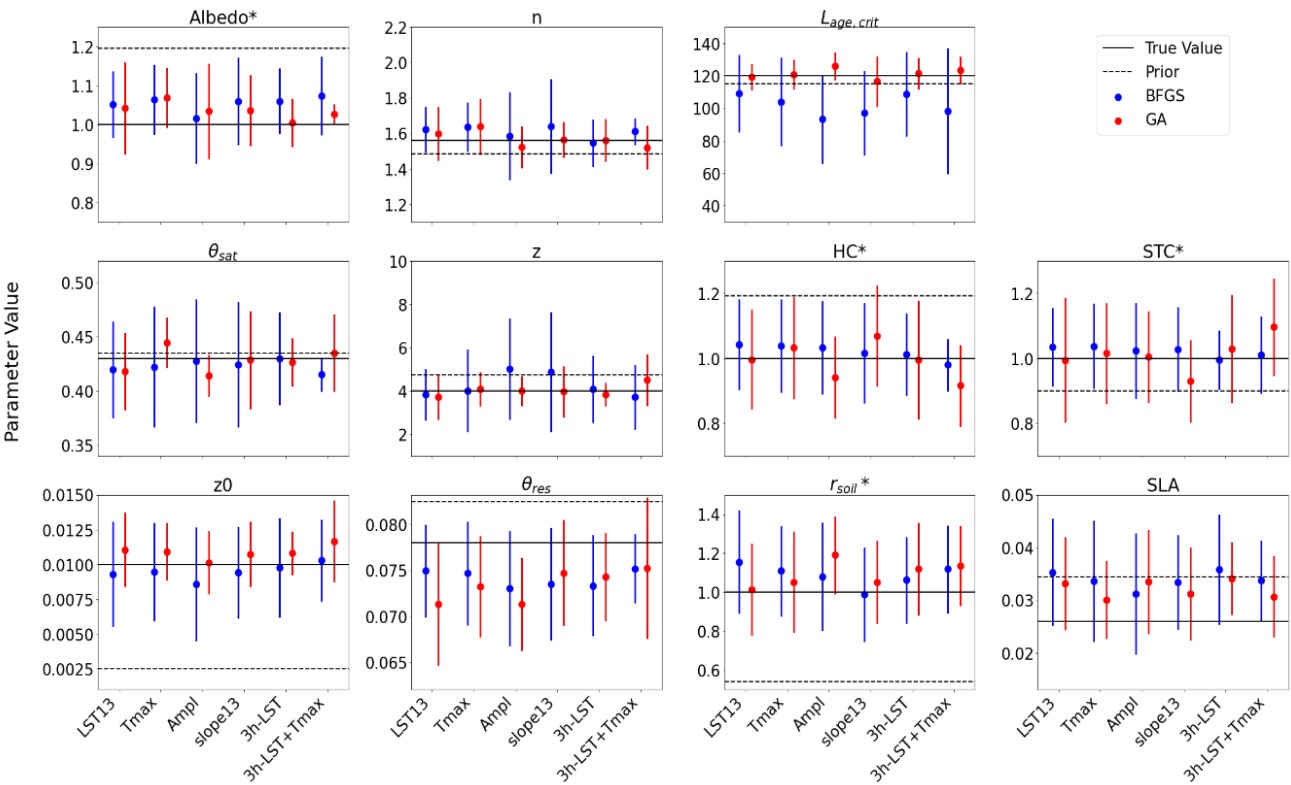

**Figure 3. Parameter estimates for each DA experiment over the selected site in Spain (ES-Abr) comparing the 11 optimized parameters between the gradient-based (in blue) and genetic (in red) methods. Parameter estimates are represented by the mean and standard deviation across 16 optimization tests with random first-guess parameter values. The x-axis indicates the DA experiment assimilating a single characteristic of the diurnal cycle (LST13, Tmax, Ampl and slope13), the 3-hourly LST pseudo-observations (LST) and including the Tmax (LST+Tmax). The 'true' parameter (default ORCHIDEE value) and prior values (defined randomly) are represented by the solid and dashed horizontal lines, respectively.**

In our efforts to evaluate the potential of assimilating the CCI-LST product to constrain ORCHIDEE parameters using twin experiments, we also consider the actual availability and uncertainties associated with CCI-LST over the pixel corresponding to the ES-Abr site (Appendix C). We conduct two DA experiments with the GA method: assimilating the 3-hourly LST series alone (3h-LST DA) and incorporating the Tmax (3h-LST+Tmax DA). For both DA experiments, the RMSD values are comparable to those obtained when considering the full pseudo-data series, although lower, particularly for LE (Fig. C1 in Appendix). However, the availability of CCI-LST data may significantly impact the estimation of daily maximum LST (Tmax) − as well as the other characteristics − especially in sites characterized by climates with higher cloud occurrence compared to the ES-Abr site . Consequently, we will conduct the real DA experiments by only considering the entire 3-hourly CCI-LST series (LST DA).

## 3.2 DA based on actual LST observations

From the above twin experiments, we concluded that the optimal DA setup to be used with the CCI-LST product is the assimilation of the 3-hourly CCI-LST data with the GA method. We thus run such optimization for the year 2018 across the 34 WarmWinter sites. Figure 4 illustrates the annual and diurnal cycles of LST in June at two contrasting sites: Chamau in Switzerland (CH-Cha, Fig. 4a-b) characterized by a humid and cool temperate climate (Cfb, according to Köppen-Geiger), and Las Majadas South in Spain (ES-LM2, Fig. 4c-d) with a dry Mediterranean climate (Csa). LST simulations using default ORCHIDEE parameters (Prior) and optimized parameters after assimilating CCI-LST data (Optimized) are compared against in situ fluxes. In both sites, the Prior LST simulations are clearly overestimated at both annual and diurnal cycles. The Optimized LST is improved in both sites, with the RMSD being decreased from 3.6 K to 2.2 K in CH-Cha and from 2.8 K to 2.2 in ES-LM2 on a daily basis. The improvement is more noticeable at hourly time scale in daytime hours during June (Fig. 4b,d), particularly in the CH-Cha site, where the assimilation of CCI-LST data significantly corrects the Prior overestimation. In the CH-Cha site, the RMSD is decreased from 5.2 K to 1.8 K, while in the ES-LM2 the RMSD is decreased from 3.1 K to 2.0 K in ES-LM2. Although the bias in Prior LST on a monthly scale is substantially corrected, the assimilation process encounters difficulties in fully addressing the overestimation observed in winter months (Fig. 4a-c). The latter is mainly attributed to a reduced availability of data during colder months. It is noteworthy that the number of observations in winter months can be nearly a third of that in summer, contributing to the difficulties observed. In addition, the issue is also influenced and inherently connected to a weakened constraint of LST on surface energy balance under radiation-limited conditions.

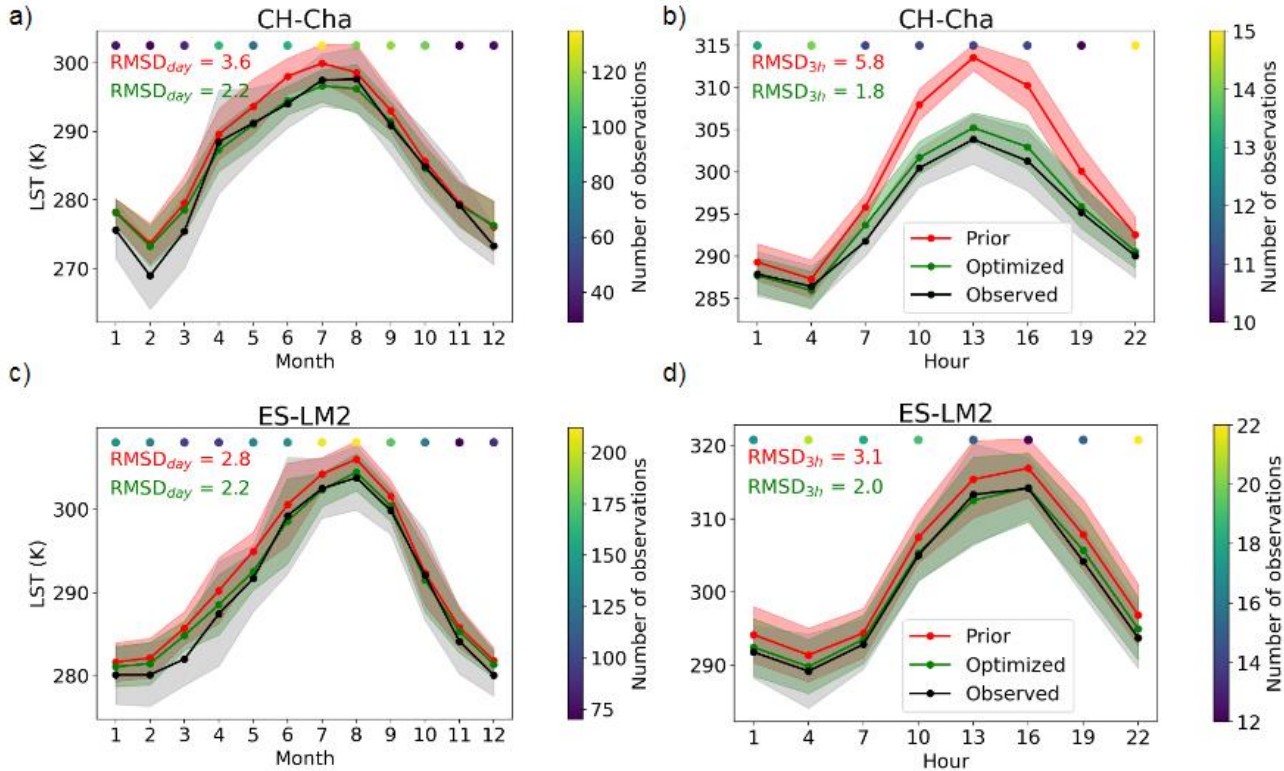

**Figure 4. Annual (a, c) and diurnal cycle in June (b, d) of LST for 2018 over a grassland (CH-Cha: a, b) and a cropland site (ES-LM2: c, d). Observed LST represents the mean (black dot) and standard deviation (shaded area) of 3-hourly CCI-LST data. Mean and standard deviation of Prior and Optimized LST are computed from the same CCI-LST availability. Colored dots at the top of each plot represent the number of CCI-LST observations per month or hour. The RMSD on the daily (RMSD_day in a, c) and half-hourly (RMSD_3h in b, d) basis is shown for Prior (red) and Optimized simulations. Note that the half-hourly LST is evaluated at 3-hourly intervals when CCI-LST is available.**

Figures 5 and 6 illustrate the annual and diurnal (in June) cycles of Prior and Optimized Rn, LE, and H compared against in situ observations at the same contrasting sites (CH-Cha and ES-LM2). At both sites, Prior Rn and LE are underestimated, while Prior H is overestimated, which is linked to the overestimated Prior LST shown in Fig. 4. Similarly to the improvements observed in LST simulations, the assimilation of CCI-LST data effectively corrects much of the underestimation in Rn and LE at both monthly (Fig. 4a, c) and hourly scales (Fig. 4b, d). Notably, improvements are evidenced during summer months and daylight hours (between 08:00 and 18:00 UTC). At both sites, the LE fluxes exhibit more pronounced improvements compared to Rn and H, in contrast to the twin DA experiments where the improvement is more important for H. At the CH-Cha site (Fig. 5), the RMSD values in LE fluxes are reduced from 43.8 W/m² to 18.5 W/m² and from 122.2 W/m² to 47.3 W/m² on a daily and half-hourly basis, respectively, both representing an improvement of about 60%. While at the ES-LM2 site (Fig. 6), the RMSD values are reduced from 20.1 W/m² to 8.8 W/m² and from 62.2 W/m² to 35.3 W/m² on a daily and half-hourly basis, respectively. Regarding H, significant enhancements are observed at CH-Cha, whereas improvements in ES-LM2 are slight and mostly noticeable from March to May and during nighttime hours (between 19:00 and 06:00 UTC). In fact, the RMSD in half-hourly H at CH-Cha is decreased from 63.5W/m² to 29.4 W/m², while at ES-LM2 it is slightly increased from 60.3 W/m² to 63.5 W/m². The disparity in improvements between the two sites can be attributed to the complexity of surface heterogeneity. Additionally, soil heat fluxes and water infiltration capacity, both presenting significant uncertainties in modeling and measurement, become particularly crucial in semi-arid sites. For instance, at the ES-LM2 site, H is significantly overestimated

during daytime hours in summer, indicating that the soil heat flux is correspondingly underestimated. CH-Cha, characterized as a homogeneous grassland site and well-classified in ORCHIDEE's PFT maps, experiences significant improvement in all three surface energy fluxes. Conversely, ES-LM2, a more complex savanna site predominantly covered by *Quercus Ilex* and grass but classified as croplands when used in ORCHIDEE, shows a comparatively smaller improvement, particularly in terms of H.

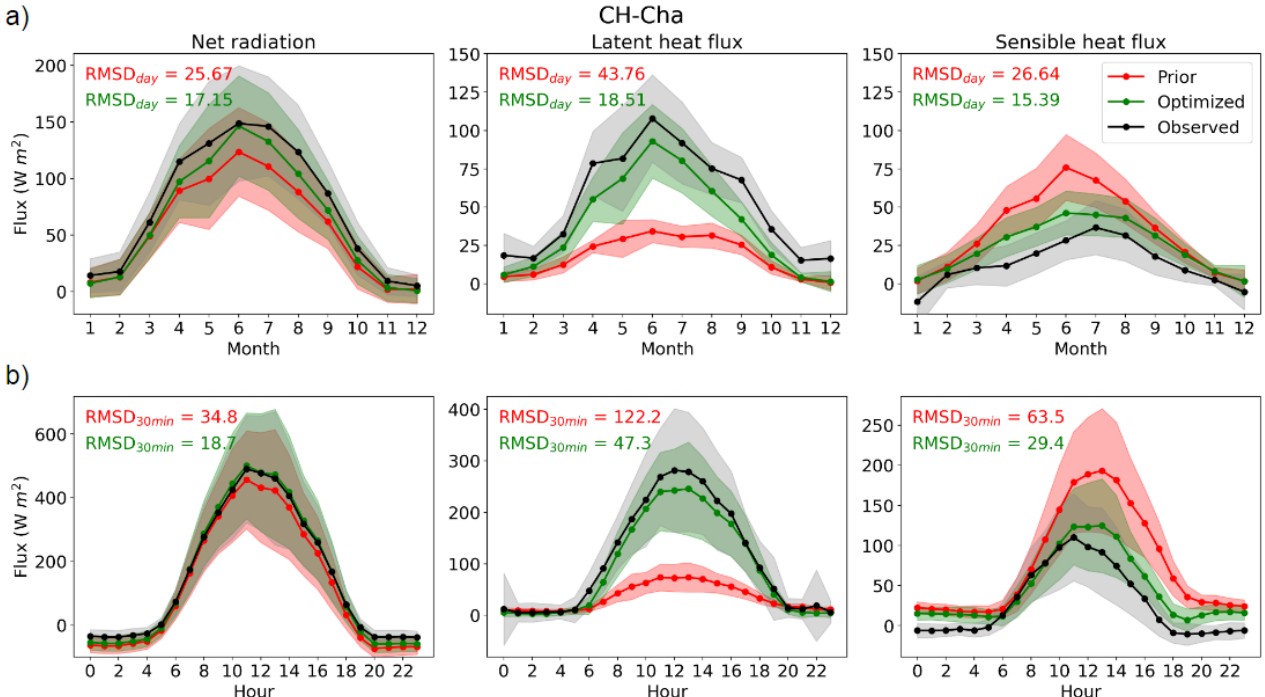

**Figure 5. Annual (a) and diurnal cycle in June (b) of Rn (left), LE (middle) and H (right) for 2018 over a grassland (CH-Cha) site. The mean (dot) and standard deviation (shaded area) are represented for in situ observations (black), Prior (red) and Optimized (green) ORCHIDEE simulations. The RMSD on the daily (RMSD$_{day}$ in a) and half-hourly (RMSD$_{30min}$ in b) basis is shown for Prior (red) and Optimized simulations.**

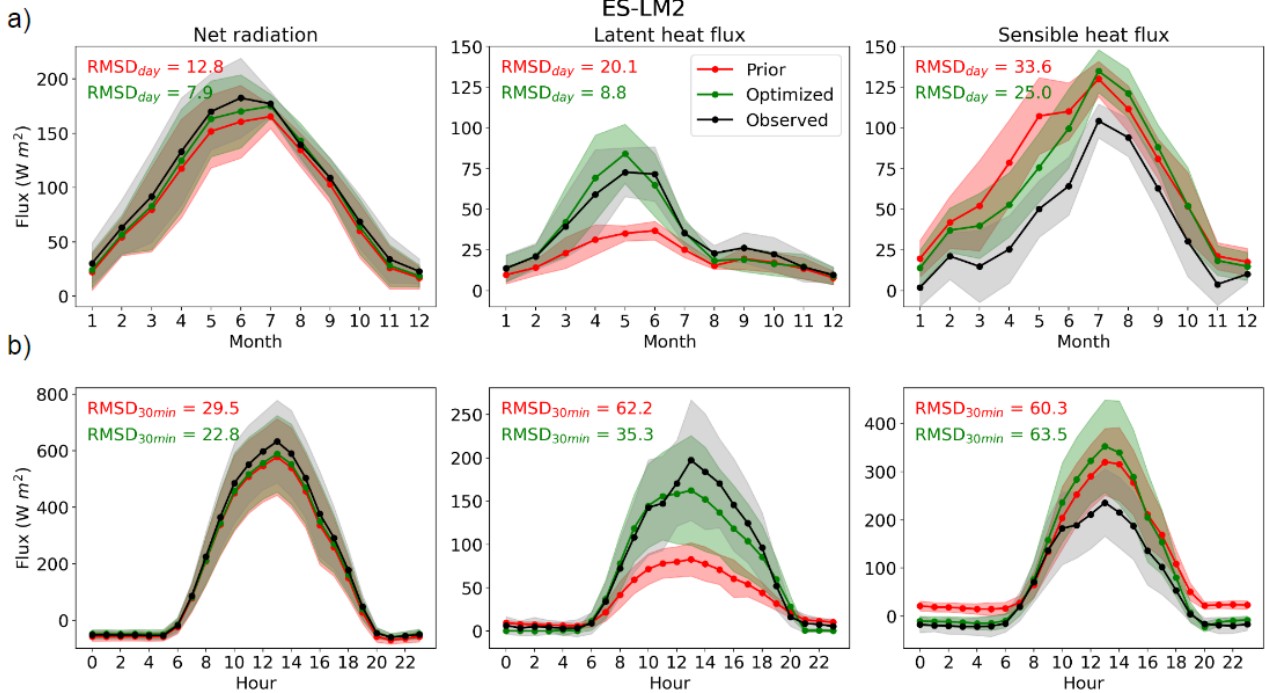

**Figure 6. Annual (a) and diurnal cycle in June (b) of Rn (left), LE (middle) and H (right) for 2018 over a cropland site (ES-LM2). The mean (dot) and standard deviation (shaded area) are represented for in situ observations (black), Prior (red) and Optimized (green) ORCHIDEE simulations. The RMSD on the daily (RMSD_day in a) and half-hourly (RMSD_30min in b) basis is shown for Prior (red) and Optimized simulations.**

Figure 7 illustrates the performance of assimilating 3-hourly CCI-LST data for the year 2018 across the 34 sites, showcasing the RMSD reduction in LST, Rn, LE, and H. Throughout all the sites, the assimilation of CCI-LST data leads to improvements in LST simulations. The enhancements in LST reach up to 60% of RMSD reduction from Prior to Optimized simulations with a median value of 24.6% across all sites. Furthermore, in the majority of the sites, the assimilation also yields improvements in the three energy fluxes. Specifically, Rn, LE, and H exhibit RMSD reduction improvements in 72.0%, 79.4%, and 70.6% of the sites, respectively. Remarkably, the most substantial enhancements are observed in LE, with RMSD reduction up to 60% and median values of 19.9%, followed by Rn and H, both with medians of 9.5%.

We observe that superior performances in LST and turbulent fluxes (LE and H) are achieved in grassland and cropland sites, while the worst results are observed in evergreen needleleaf forest (ENF) sites. Particularly noteworthy are the substantial improvements in LE for grassland sites, with median RMSD reduction per PFT ranging between 21% and 47%. Meanwhile, H improvements in grassland and croplands are about 20%. In contrast, the assimilation outcomes in forest sites do not demonstrate comparable success. It is important to highlight that the four ENF sites with cool boreal climates show degradation in fluxes after assimilation, with the exception of the CH-Aws site. This deterioration is linked to the complex terrain of these four sites, situated in mountainous regions with a cool boreal climate, including three in the Alps (CH-Aws, CH-Dav, and IT-Ren) and one in the Carpathians (CZ-BK1). On the other hand, the ENF in other climates demonstrates varied performances across sites, with no overall changes in RMSD for LE (median RMSD reduction of 0.9%). However, there is an improvement in terms of RMSD reduction for H, approximately 9.3%. That can be explained by the fact that evapotranspiration in these forests is less affected by water stress as their roots can extract water deeper, and therefore the vegetation temperature is more stable and doesn't contribute much to the optimization of LE.

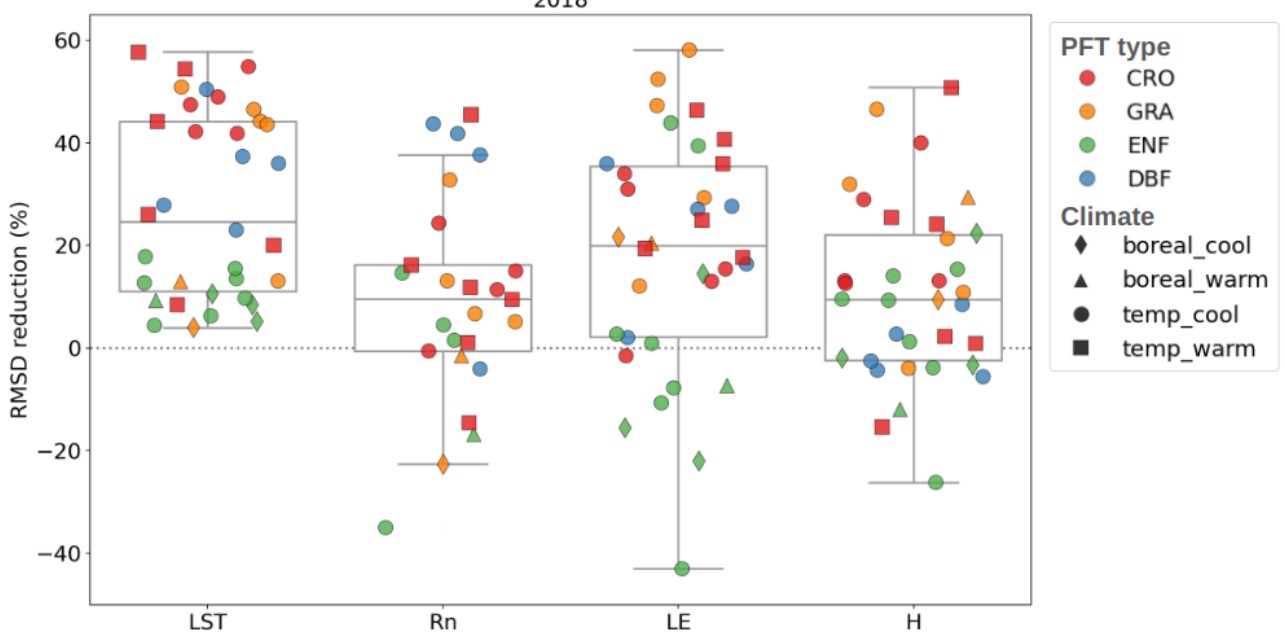

**Figure 7. Boxplots showing the performances in terms of model–data RMSD reduction (%) obtained for hourly Rn, LE and H over sites assimilating 3-hourly LST from the ESA-CCI product in 2018. Note that in 2018, Rn is available in only 25 of the 34 sites assessed.**

Figure 8 depicts the breakdown of MSE into bias (SB), difference of standard deviations (SDSD) and lack of correlation (LCS) across sites for LST and the three simulated fluxes using the default ORCHIDEE parameters (Prior) and the optimized parameters after assimilating CCI-LST data (Optimized). The MSE in LST is significantly decreased from Prior to Optimized ORCHIDEE simulations in terms of both median and spread values. In practice, the median MSE decreases from 3.91 K to 2.50 K, representing an MSE reduction of 36% from Prior to Optimized LST in model-data fit. This MSE reduction comes mainly from the reduction of the two major components of Prior MSD: SB and LCS. The bias component (SB) shows an important improvement that is decreased from 2.20 K to 0.49 K (i.e. decreased by 78%) from Prior to Optimized. Although the SDSD component shows a degradation (increased) after assimilation, this error remains small (i.e. with median values from 0.77 K to 1.06 K). This degradation is explained by a slight underestimation of the standard deviation of simulations compared to that of observations, which are the first and second terms of Eq. (5), respectively. Consequently, this leads to an augmentation in the disparity between the standard deviations of simulations and observations.

The MSE components in Rn show similar and relatively smaller values, ranging between 0 and 40 W/m², whereas in LE and H they can exceed 80 W/m². After assimilation, significant improvements are evident across all three components (SB, SDSD, and LCS), as observed in the medians of Optimized simulations. With the exception of LCS in LE, the medians are significantly reduced for the three fluxes, reflecting a substantial reduction in the MSE components after assimilating CCI-LST data. For LE, the ORCHIDEE model using default parameters (Prior) struggles mainly in simulating the amplitude of the seasonal cycle across a majority of sites, as evidenced by the substantial Prior SDSD with a high median ($35.5$ W/m$^2$) and significant spread. SDSD is significantly reduced by 70% after assimilating LST data, along with a 40% reduction in the bias component (SB). However, the fluctuations pattern is compromised, as indicated by a larger LSC for Optimized simulations, signifying a deterioration by 33% in the temporal pattern. Regarding H, all three components show improvement, with particular enhancements in the bias (SB) that is reduced by 40%, as for LE. Even though the temporal pattern (LCS) of H is reduced by 9% it remains the most important component for a majority of sites, as seen by a large median and spread in Optimized H.

That can be also observed in the MSE decomposition by site illustrated in Fig. D1 of the Appendix section. Therefore, the assimilation faces challenges primarily in enhancing the temporal pattern of turbulent fluxes, where LCS is the major MSE component in LST, LE and H, constituting 47%, 71% and 61% of the total MSE for LST, LE and H, respectively.

The challenge related to the temporal pattern of turbulent fluxes, represented by the lack of correlation component, is further highlighted when examining the correlation coefficient between simulations and observations, analyzed separately for each hour and month (not shown). During daytime hours (between 08:00 and 16:00 UTC), median correlations are equal to 0.81 for both LE and H. However, during nighttime hours (between 20:00 and 04:00 UTC), these correlations drop to 0.15 for LE and 0.46 for H as could be expected given the much lower values of the fluxes encountered during night time. While the disparity is less pronounced on a monthly scale, median correlations in December and January are notably lower (0.41-0.46 for LE and 0.53 for H) compared to the rest of the year. The median correlation values range between 0.64 (in November) and 0.90 (May-June) for LE and between 0.75 (November) and 0.92 for H (August).

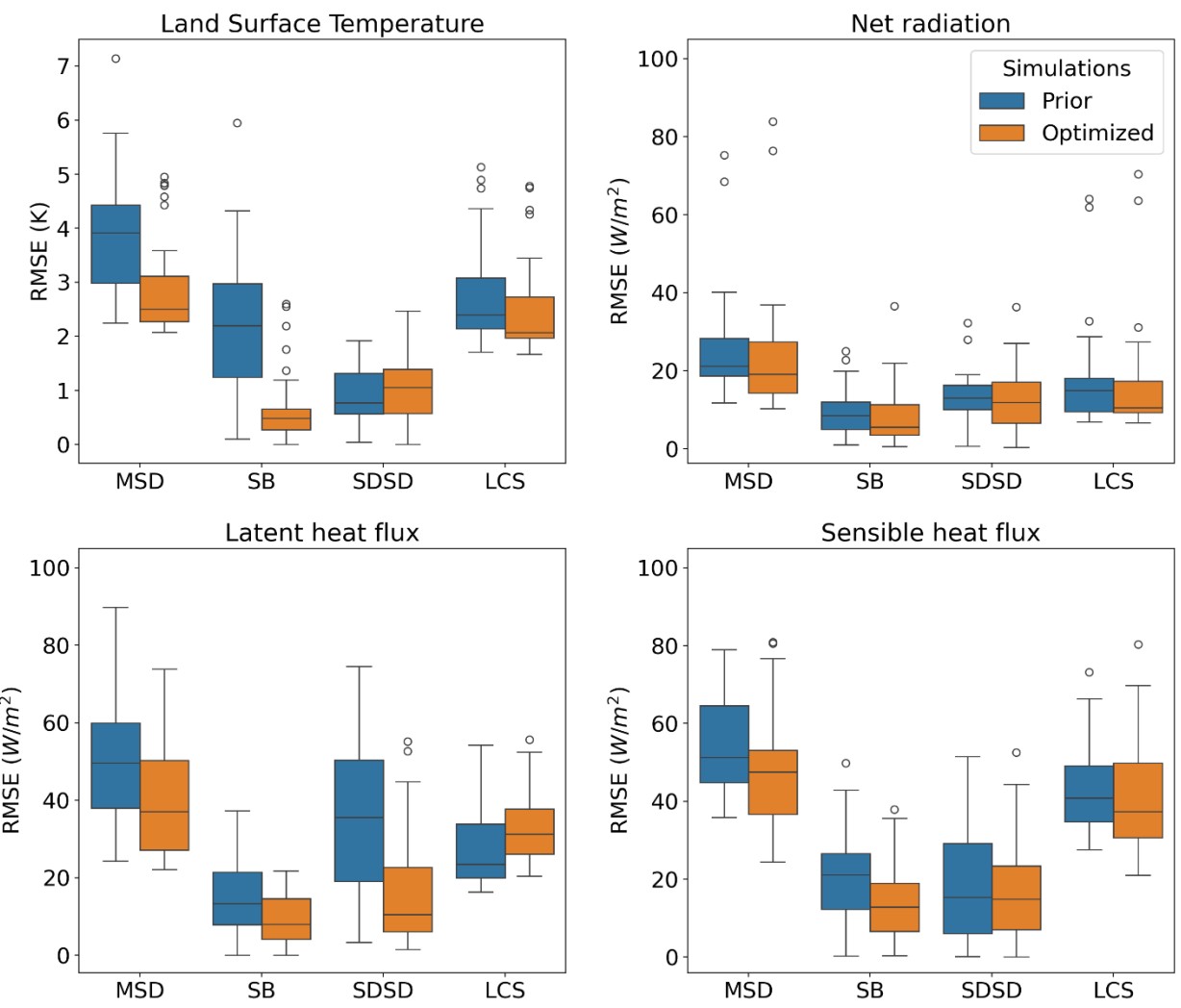

**Figure 8. Boxplots showing the decomposition of the mean square error (MSE) in terms of bias (SB), difference of standard deviations (SDSD) and lack of correlation (LCS) between the model and observations for half-hourly LST**

**(top left), Rn (top right), LE (bottom left) and H (bottom right) across sites in 2018. For clarity, note that the square root of the error components is plotted.**

Figure 9 shows the breakdown of MSE in LST per season, highlighting the periods when the assimilation of CCI-LST data exerts a more pronounced impact throughout the year. Similar to the overall MSE in LST shown in Fig. 8, the MSE in spring (AMJ in Fig. 9) and summer (JAS) experiences a substantial reduction in terms of their median values from 4.2 to 2.3 K and 3.7 to 2.3 K, respectively. Conversely, the MSE in winter (JFM) and autumn (OND) are slightly reduced from 3.0 K to 2.8 K for both seasons. In all seasons, the bias component is the most reduced component after assimilating the 3-hourly CCI-LST. For instance, during warmer seasons (AMJ and JAS), the bias component experiences the most significant reduction, with median values decreasing by 72% (from 3.0 K to 0.8 K) and 86% (from 2.2 K to 0.3 K) in AMJ and JAS, respectively. In addition, the spread of the bias component is markedly diminished during these seasons. However, in both seasons, the difference in standard deviations (SDSD) increases, indicating that the assimilation of CCI-LST data encounters challenges in improving the amplitude of the seasonal cycle, which is consistent with observations for the rest of the year. As discussed above, the smallest improvements in LST simulations in colder seasons can be mainly linked to the reduced availability of data (more cloudy conditions) during colder months, as observed in Fig. 4 for two contrasting sites. It is worth noting that in winter the availability of data is on average 63 (± 30) observations per month across all sites, while in summer the average is increased to 118 (± 35) observations per month.

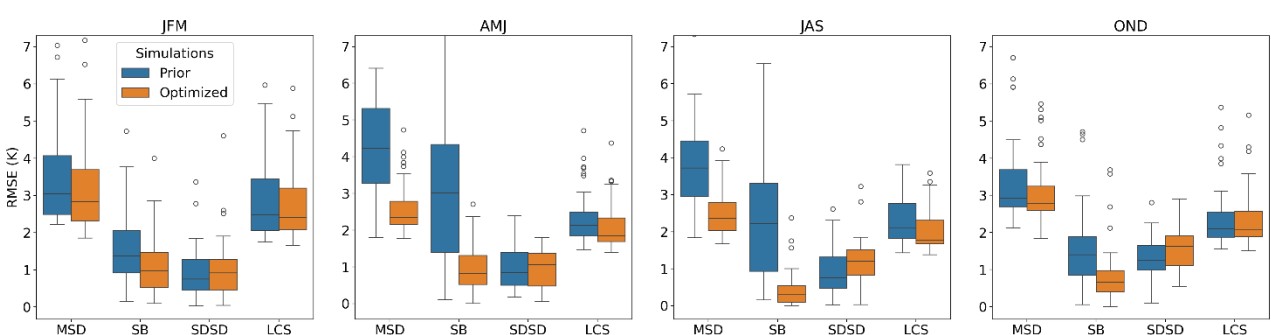

**Figure 9. Boxplots showing the decomposition of MSE per season in terms of bias (SB), difference of standard deviations (SDSD) and lack of correlation (LCS) between the model and observations for half-hourly LST across sites in 2018. For clarity, note that the square root of the error components is plotted.**

As observed in the case of LST, and closely associated with it, similar seasonal enhancements are evidenced for the surface energy fluxes (Rn, LE, and H), indicating further improvements during warmer seasons (see supplementary Fig. S4). During spring (AMJ) and summer (JAS), significant reductions in MSE are observed for all three fluxes after optimization, particularly for LE, where the MSE is reduced by 28% and 22% for AMJ and JAS, respectively. Notably, during these seasons (AMJ and JAS), LE presents significant improvement in both the bias and SDSD components. Similarly, the bias component is significantly improved for H after assimilating CCI-LST data. However, the three fluxes exhibit slight enhancements in MSE during winter (JFM) and autumn (OND), with median values of their components being quite comparable between Prior and Optimized simulations. Confirming the evidence presented in Fig. 8 across all variables assessed (LST and fluxes) during 2018, the lack of correlation component (LCS) in Optimized simulations emerges as the primary component in MSE for all seasons, exhibiting the least improvement following assimilation of CCI-LST data. Consequently, LCS remains the dominant MSE component for all variables and seasons, with the exception of Rn in spring (see supplementary Fig. S4).

### 3.2.1 Model performance on the evaluation period

The parameters optimized from the assimilation of CCI-LST data in 2018 over each site are used to simulate LST and surface energy fluxes during the evaluation period (2009-2020) for stations with available in situ data. Figure 10 shows the performances in RMSD reduction in LST, Rn, LE, and H considering optimized parameters across the 34 sites from 2009 to 2020. In contrast to the optimization year (2018), some deterioration is observed in LST for a couple of sites during the validation period. The results obtained for the three fluxes along the years are very similar to those obtained in the optimization

520    year with most of the sites presenting improvements using the optimized parameters in 2018. In fact, the medians in RMSD reduction per year are quite stable across years for the three fluxes, which are equal to 7.2%, 19.7% and 9.5% for Rn, LE and H, respectively. As found in 2018 for LE and H, better performances are evidenced in grassland and cropland sites, while the worst performances are found in ENF and boreal climates when compared to the other vegetation and climate types. Specifically, the ENF sites systematically exhibit less improvement or even deterioration for LST and LE, with RMSD

reductions falling below their median values.

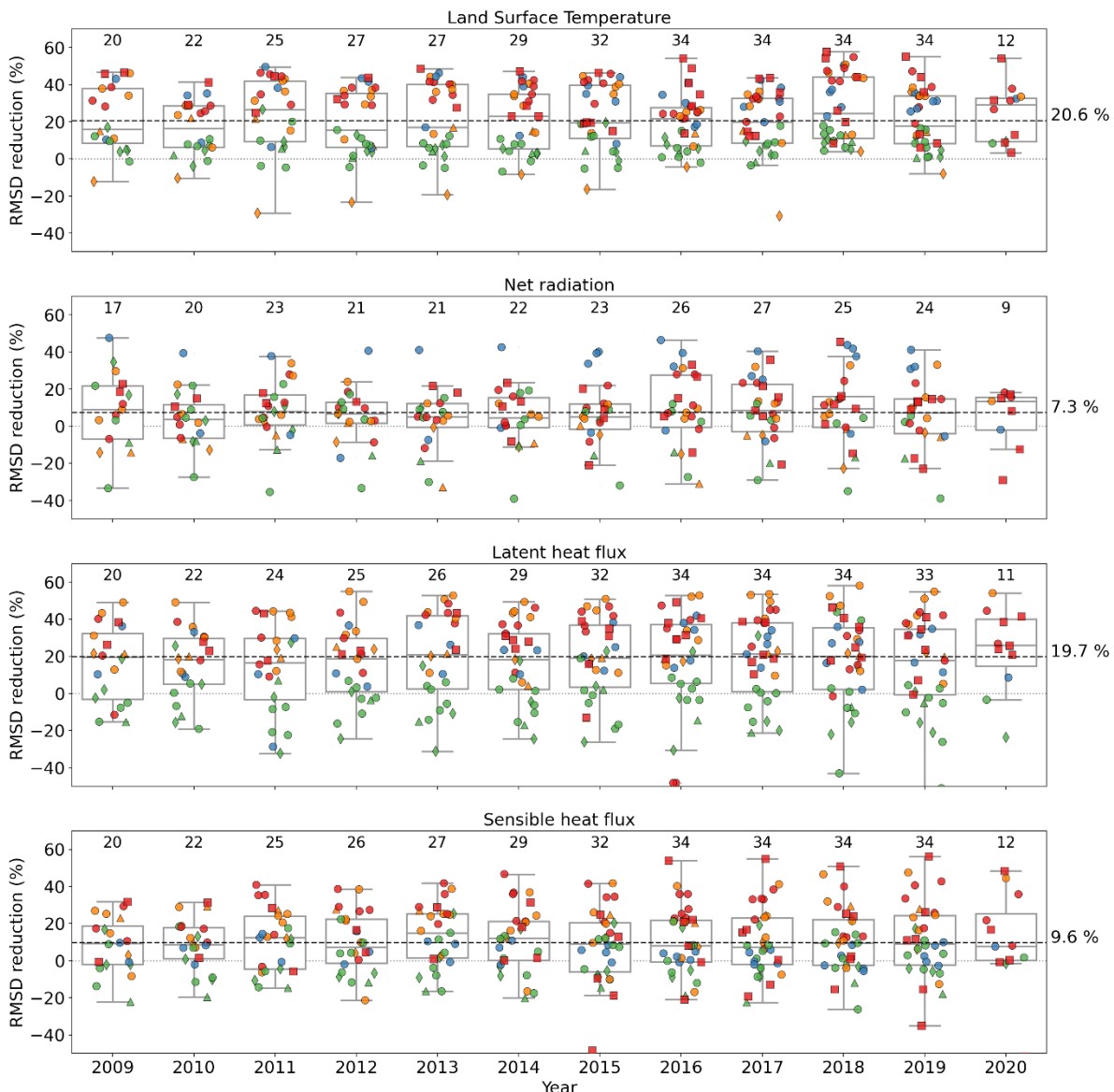

**Figure 10. Boxplots showing the performances in terms of model–data RMSD reduction (%) obtained for half-hourly LST, Rn, LE and H from 2009 to 2020 using optimized parameters per sites in 2018. The number of sites with available data per year is shown above each boxplot. The dashed black line represents the average of the medians over the years. The color represents the dominant vegetation type per site, such as CRO: red, GRA: orange, DBF: blue and ENF: green. The symbols represent the climate: warm temperate: squares, cool temperate: circles, warm boreal: triangles and cool boreal: diamond.**

### 3.2.2 Improvement in fluxes from average parameters

We calculate the median of optimized parameters across sites to account for a unique set of PFT-specific parameters. Thus, we run ORCHIDEE with this unique parameter set per PFT over the 34 sites from 2009 to 2020 to evaluate the effectiveness of optimized parameters at larger scales. This assessment aims to determine whether the RMSD reduction performances at each site align with those achieved using site-specific optimized parameters, as shown in Sect. 3.2.1.

Although the performances over the years are inferior for LST to those obtained using optimized parameters per site, the results obtained for the three fluxes remain quite similar (see Fig. 11). The averaged median in RMSD reduction across years for LST decreases to 13.8% using PFT-specific parameters, instead of 20.6% using the parameters optimized per site. In turn, the medians in RMSD reduction per year for the three fluxes exhibit not only consistent stability across the years, comparable to using site-specific parameters (Fig. 10), but also a slight increase. The average medians for Rn, LE, and H are notably higher, reaching 13.1%, 20.7%, and 9.6%, respectively, compared to average medians of 7.3%, 19.7% and 9.6% using site specific parameters. It is noteworthy that Rn experiences additional improvement when utilizing a unique set of parameters. This enhanced improvement is mainly visible in grassland and cropland sites, which show more significant improvement when using a unique set of parameters, as illustrated in Fig. 11. Particularly remarkable are the substantial enhancements in grassland sites, with all sites experiencing an improvement in Rn simulations using PFT-specific parameters compared to site specific parameters. In fact, the median RMSD reduction in Rn simulations is increased from 4% using site-specific parameters to 17% using PFT-specific parameters. These improvements are particularly observed in sites characterized by boreal climates, where the assimilation of CCI-LST data struggles to enhance simulations. However, certain sites experience a decline when employing a unique set of parameters, notably the ENF sites. The LST and Rn simulations over these sites deteriorate in comparison to both prior simulations and those utilizing site-specific optimized parameters.

In contrast, improvements in RMSD reductions for LE and H using a unique set of optimized parameters are comparable, or even superior, to using site-specific optimized parameters across the four vegetation types. Notably, grassland sites exhibit significant additional enhancements for LE and H when employing a unique set of parameters, with the exception of sites under temperate cool climates for LE.

While multi-site optimization using only LE and H data has proven further enhancements in model performance compared to averaging site-specific parameter values (Kuppel et al., 2012), its implementation can become challenging with LST data, given that we have numerous PFTs and soil textures, resulting in a multitude of cases to consider. Our findings suggest that using the median of site-specific parameters can offer a practical and effective alternative for calibration, particularly in cases where a multi-site setup would be overly complex.

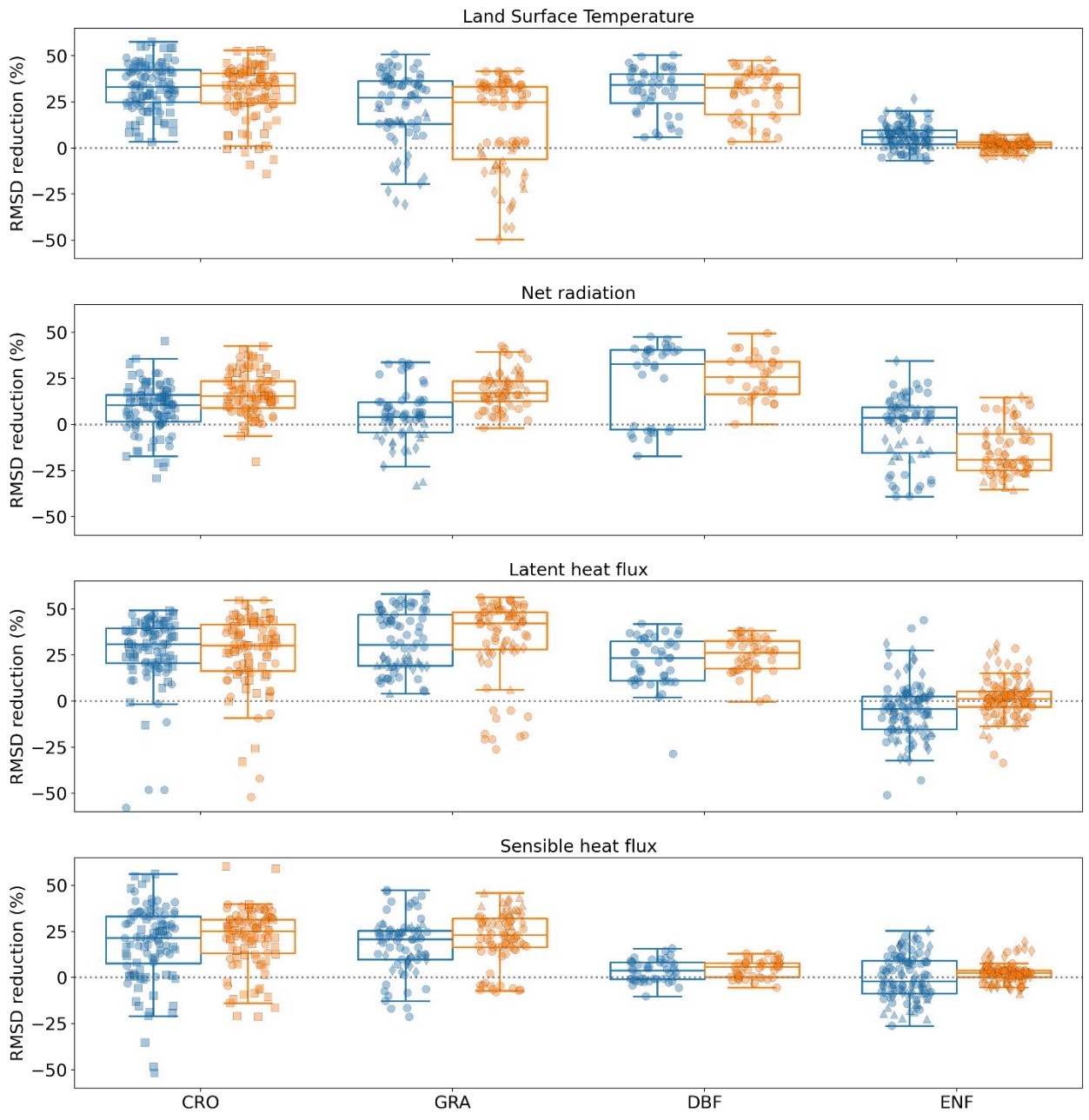

**Figure 11. Boxplots showing the performances in terms of model–data RMSD reduction (%) obtained for half-hourly LST, Rn, LE and H across all sites from 2009 to 2020 using optimized parameters per site (blue) and a single set of parameters per PFT (orange). The site-specific parameters are obtained from the assimilation of 3-hourly CCI-LST observations in 2018, while the PFT-specific parameters are obtained from the median of these site-specific parameters per vegetation type.**

**4 Discussion**

According to our results, we expect that using the PFT-specific parameters derived from the 34 evaluated sites will enhance ORCHIDEE simulations of LST and energy fluxes on a regional scale. While improvements are anticipated for croplands, grasslands, and deciduous broadleaf forests, they are less likely for evergreen needleleaf forests, especially in boreal climates.

It is important to note that in this study, only 7 out of the 15 ORCHIDEE PFTs are represented by these sites, with some, like boreal evergreen needleleaf forests, being underrepresented. This study seeks to improve the simulations of LST and surface energy fluxes on a regional scale. However, ongoing work aims to assimilate CCI-LST data across various pixels for all PFTs, ensuring identical representation (i.e., the same number of pixels per PFT).

## 4.1 Impact of the optimization period

We acknowledge that the selection of a particular year of optimization may impact the selection of parameters to optimize and the performance of the assimilation. Regarding the selection of parameters for optimization, we previously conducted two sensitivity analyses at the 34 sites for both 2017 and 2018, separately. The selected parameters for optimization were generally consistent between the two years, with the exception of the parameters controlling the water stress curve ($\alpha$) and the critical soil moisture above which transpiration is maximal ($\theta_{crit,rel}$). In 2017, $\alpha$ and $\theta_{crit,rel}$ were selected for optimization at 7 and 6 sites, respectively, whereas in 2018, both parameters were selected at 13 sites. This difference is attributed to the drought conditions in 2018, which increased the relevance of these water stress-related parameters. Properly representing these parameters is crucial for future projections of climate and water resources (Fu et al., 2022; Fu et al., 2024), highlighting the importance of considering appropriate conditions for accurately optimizing the processes we aim to improve.

In the twin experiments at the ES-Abr site, we previously assessed the impact of selecting a specific year (2018) versus the entire available 6-year period (2015-2020) on the performances of half-hourly LST and turbulent fluxes during 2017 (see Fig. E1 in Appendix E). We chose 2017 to ensure a more independent evaluation of both calibration periods (2018 and 2015-2020). The results showed no significant differences in improving the fluxes in 2017 between using the entire period (2015-2020) and a single year (2018) for calibration. Although using the entire period resulted in a slightly higher RMSD reduction for the three variables with the GA method, the BFGS method yielded superior performances when using only 2018 for calibration. This may seem counterintuitive, as additional information typically creates extra constraints, helping to smooth the cost function and making local minima less likely for BFGS. However, our findings can be explained by the fact that drought periods in 2018 are less predominant in a 6-year period, resulting in a less optimal solution for the calibration of the water stress parameters with the BFGS method.

## 4.2 Impact on vegetation phenology and soil water

We recognize that assimilating LST alone has its limitations and cannot enhance the model-data fit of all variables controlling water, energy, and carbon fluxes. To better understand the performance of the LST assimilation procedure on other variables less directly linked to energy fluxes, we assessed the impact of LST optimization on soil water availability and gross primary productivity (GPP). Since the number of sites with available soil moisture data are limited and measurement depths vary among sites, we evaluated the impact on the top 10 cm of soil moisture in our twin experiments. The soil moisture showed a clear improvement (positive median RMSD reduction) when assimilating the 3-hourly LST alone using the GA method. The median RMSD reduction for this experiment (3h-LST) represented an enhancement in soil moisture of 10.4%, although some runs among the 16 different first-guesses resulted in a deterioration of soil moisture. The fact that the 3h-LST DA showed an overall improvement in soil moisture confirms the chosen strategy for the assimilation of the CCI-LST data.

Regarding the GPP, we assess the impact by assimilating the 3-hourly CCI-LST time series over the 34 WarmWinter sites in 2018. Assimilating the 3-hourly LST data results in an overall degradation in GPP, with a median RMSD increase across sites of 7.4%. Among the studied sites, 14 out of 34 shows improvements in diverse conditions such as the grassland CH-Cha and Mediterranean ES-LS2 sites (see Fig. F1 in Appendix F). At the other sites, larger errors were obtained, with RMSD increases of up to 55.4%. For instance, while the cropland CH-Oe2 site exhibits a 36.1% improvement in LE, the RMSD in GPP is increased by 41.3% (see Fig. F2 in Appendix F). Despite the mixed results obtained on GPP, the improvement observed in the

14 sites (i.e., 41% of the sites) is a promising outcome, especially considering the challenge of enhancing model variables that are not closely linked to LST. In fact, assimilating a single data stream may even degrade the model simulations of other variables, as shown in Kato et al. (2013) and Bacour et al. (2015; 2023). In our study, since we calibrated only parameters impacting LST and kept the carbon-related parameters previously optimized without LST observations, a degradation of the carbon fluxes is not surprising. Nonetheless, the overall improvement of the energy fluxes, such as the 20.6% enhancement in LE and 9.6% in H, is significantly more impactful than the observed degradation in GPP.

Although improvements in soil moisture and phenology were not expected by assimilating only LST data, the enhancements found in the twin experiments for soil moisture and some sites for GPP with actual data are very encouraging. These results support ongoing efforts to jointly assimilate LST with satellite-derived products such as leaf area index, albedo or soil moisture into ORCHIDEE. Such an approach is expected to better constrain a wider range of energy, water, and carbon parameters, enhancing the overall performance of the model.

## 5 Summary and Conclusions

This study focuses on the assimilation of ESA-CCI LST data into the ORCHIDEE land surface model with the aim of refining LST and surface energy fluxes simulations. Through a series of synthetical twin DA experiments, we explore different optimization methods (BFGS and GA) and assimilated state variables (individual LST observations and characteristics of the LST diurnal cycle) to determine the most effective assimilation strategy. The selected strategy is then implemented to assimilate actual CCI-LST data across 34 European sites in an optimization year (2018) and, finally, validated from 2009 to 2020.

The results from the twin DA experiments reveal that the genetic algorithm (GA) consistently outperforms the BFGS method, evidencing more substantial improvements than those achieved by BFGS in all DA experiments for both LST and turbulent fluxes. This superiority is underscored by the reliability and consistency exhibited by the GA method. This is exhibited not only in RMSD reductions from prior to posterior simulations but also in the consistent convergence to the 'true' parameter values across the 16 runs with random first-guesses, particularly for the most LST-sensitive parameters.

Concerning the performances of the model optimization, our findings show that the most substantial enhancements are evidenced when considering the entire 3-hourly LST series, either individually (LST DA experiment) or jointly with other attributes of the diurnal cycle (LST+Tmax, LST+Ampl and LST+Ampl+Tmax DA experiments). In contrast, assimilating a single characteristic of the LST diurnal cycle (e.g. LST at 13h, daily minimum, maximum, amplitude, morning and afternoon gradients) yields comparatively smaller improvements in both LST and H simulations. Conversely for LE, assimilating a single characteristic such as LST13, Tmax, Tmin and Ampl, resulted in improvements very close to that obtained by the entire 3-hourly LST series. Our findings suggest that assimilating LST at 13:00, as will be possible with the forthcoming TRISHNA and LSTM missions, can significantly enhance LE simulations. This highlights the valuable contribution that these missions can make to the future modeling of the Earth's surface and monitoring of water resources. Nevertheless, through the combination of different characteristics of the diurnal cycle, noteworthy improvements similar to those achieved using the entire 3-hourly LST series can be reached for both LST and turbulent fluxes. It should be noted that these outcomes are obtained considering the full availability of 3-hourly pseudo-observations, so they might be considerably degraded with the actual availability of CCI-LST considering cloudy conditions.

Therefore, we proceed with the assimilation of actual CCI-LST data over 34 European sites, focusing exclusively on the complete 3-hourly CCI-LST series (LST DA). The limitation stems from the scarcity of available LST observations per day, especially in sites situated in boreal climates that are more prone to occurrences of cloudy conditions. Consequently, this

specific DA experiment is identified as the most effective assimilation strategy, as introducing additional characteristics to the LST series does not yield a substantial further advantage in the assimilation process.

The optimization of key parameters per site leads to a remarkable enhancement in the surface energy fluxes at the in situ level, with improvements observed across LST, Rn, LE and H. Notably, the optimization conducted over a single year yields improved ORCHIDEE simulations over the entire 11-year validation period. The benefits of this optimization are not uniform and vary depending on vegetation types and climates. For instance, cropland and grassland sites exhibit larger reductions in RMSD compared to forested sites, particularly in evergreen needleleaf forests where degraded simulations are observed after

assimilation. Similarly, warmer climates show greater RMSD reduction than boreal climates, where the assimilation of LST struggles to enhance LST and energy fluxes. The latter is explained by the fact that the water is not limiting in the evergreen needleleaf forest sites and under cold climates, leading to a weakened LST-evapotranspiration relationship (i.e. energy-limited evapotranspiration regimes).

To evaluate the applicability and effectiveness of optimized parameters on a broader scale, from regional to global scales, we

employ a unique set of parameters per PFT obtained from the optimization per site. We evaluate ORCHIDEE simulations using the median of parameters specific to vegetation type (PFT-specific) across all 34 sites from 2009 to 2020. Significantly, the performances for both LST and fluxes exhibit not only consistent stability over the years, comparable to using site-specific parameters, but also indicate a slight improvement in energy fluxes. However, assimilating LST alone has limitations and cannot improve all variables controlling water, energy, and carbon fluxes. Nevertheless, our findings reveal promising

outcomes, such as the clear improvement in soil moisture in the twin experiment and the enhancement of GPP at several studied sites. Despite the challenges, these results indicate that LST data can positively influence variables less directly linked to energy fluxes. This underscores the potential of combining LST with other satellite-derived products, such as leaf area index, albedo, and soil moisture, to better constrain and improve the overall performance of the ORCHIDEE model.

Furthermore, our findings underscore the notable impact of the ESA CCI-LST product and its associated uncertainty in

effectively constraining water and energy fluxes within the ORCHIDEE model. This suggests that integrating CCI-LST data can substantially contribute to further improving climate simulations with Earth System Models and advancing our comprehension of land-atmosphere interactions. However, future research is essential to refine the utilization of uncertainties provided by the CCI-LST product. This involves integrating time-varying observation errors derived from the 3-hourly LST uncertainty associated with each observation into ORCHIDAS, as well as exploiting the decomposed uncertainties while

considering their spatio-temporal variability.

**Code and data availability**

The source code for the ORCHIDEE version 2.2 used in this model is freely available online via the following address: http://forge.ipsl.jussieu.fr/orchidee, last access: 14 February 2024. The ORCHIDEE model code is written in Fortran 90 and is maintained and developed under a Subversion (SVN) version control system at the Institute Pierre-Simon Laplace (IPSL)

in France. The ORCHIDAS data assimilation scheme (in Python) is available through a dedicated website (https://orchidas.lsce.ipsl.fr, last access: 14 February 2024).

The ESA CCI-LST v1.1 product used in this study is available at https://catalogue.ceda.ac.uk/uuid/6775e27575124407afeebb4bb1dfaaf5 (Veal et al., 2022). The WarmWinter database is available at https://www.icos-cp.eu/data-products/2G60-ZHAK (Warm Winter 2020 Team and ICOS Ecosystem Thematic

Centre, 2022).

**Author contributions**

LOG, CO, NR and PP conceived the research. LOG performed simulations, conducted the analysis and validation. LOG prepared the manuscript with contributions from all co-authors. CO, NR and PP supervised the research. All co-authors reviewed the paper.

**Competing interests**

The authors declare that they have any competing interests.

**Acknowledgments**

This work was partially supported by the European Space Agency (ESA) as part of an ESA Climate Change Initiative (CCI) Fellowship (ESA ESRIN/Contract No. 4000133601) (project DESTRESS). L. Olivera-Guerra was also partly supported by 685 the European project, CORSO (grant agreement 101082194). We would like to thank Camille Abadie for processing the WarmWinter dataset used to run and evaluate the ORCHIDEE optimizations in this study. We would also like to thank Vladislav Bastrikov for developing and maintaining the ORCHIDAS code. Finally, the authors thank the reviewers for their useful comments which helped to improve the manuscript.

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

 **Appendix A - Sites and pseudo-observations assimilated**

**Table A1.** Main characteristics of sites used in this study in terms of location, availability of data (Years) vegetation and climate types.

| Site | Latitude | Longitude | Years | Vegetation type[*] | Climate type[**] |
|------|----------|-----------|-------|-----------------|----------------|
| BE-Dor | 50.31 | 4.97 | 2011-2020 | CRO | temp_cool |
| BE-Lon | 50.55 | 4.75 | 2004-2020 | CRO | temp_cool |
| BE-Maa | 50.98 | 5.63 | 2016-2019 | ENF | temp_cool |
| CH-Aws | 46.58 | 9.79 | 2006-2019 | ENF | boreal_cool |
| CH-Cha | 47.21 | 8.41 | 2005-2019 | GRA | temp_cool |
| CH-Dav | 46.82 | 9.86 | 1997-2019 | ENF | boreal_cool |
| CH-Lae | 47.48 | 8.36 | 2004-2019 | DBF | temp_cool |
| CH-Oe2 | 47.29 | 7.73 | 2004-2019 | GRA | temp_cool |
| CZ-BK1 | 49.50 | 18.54 | 2004-2019 | ENF | boreal_warm |
| CZ-Lnz | 48.68 | 16.95 | 2015-2019 | DBF | temp_cool |
| CZ-RAJ | 49.44 | 16.70 | 2012-2019 | ENF | temp_cool |
| CZ-Stn | 49.04 | 17.97 | 2010-2019 | DBF | temp_cool |
| CZ-wet | 49.02 | 14.77 | 2006-2019 | GRA | temp_cool |
| DE-Geb | 51.10 | 10.91 | 2001-2019 | CRO | temp_cool |
| DE-Gri | 50.95 | 13.51 | 2004-2019 | GRA | temp_cool |
| DE-Hai | 51.08 | 10.45 | 2000-2020 | DBF | temp_cool |
| DE-HoH | 52.09 | 11.22 | 2015-2019 | DBF | temp_cool |
| DE-Hzd | 50.96 | 13.49 | 2010-2019 | ENF | temp_cool |
| DE-Kli | 50.89 | 13.52 | 2004-2019 | CRO | temp_cool |
| DE-Obe | 50.79 | 13.72 | 2008-2019 | ENF | temp_cool |
| DE-RuR | 50.62 | 6.30 | 2011-2020 | GRA | temp_cool |
| DE-RuS | 50.87 | 6.45 | 2011-2020 | CRO | temp_cool |
| DE-RuW | 50.50 | 6.33 | 2012-2020 | ENF | temp_cool |
| DE-Tha | 50.96 | 13.57 | 1996-2019 | ENF | temp_cool |
| ES-Abr | 38.70 | -6.79 | 2015-2020 | CRO | temp_warm |
| ES-LM1 | 39.94 | -5.78 | 2014-2020 | CRO | temp_warm |
| ES-LM2 | 39.93 | -5.78 | 2014-2020 | CRO | temp_warm |
| FR-Lam | 43.50 | 1.24 | 2005-2019 | CRO | temp_warm |
| IT-BCi | 40.52 | 14.96 | 2004-2020 | CRO | temp_warm |
| IT-Lav | 45.96 | 11.28 | 2003-2019 | ENF | temp_cool |

| | | | | | |
|---|---|---|---|---|---|
| IT-Lsn | 45.74 | 12.75 | 2016-2020 | CRO | temp_warm |
| IT-MBo | 46.01 | 11.05 | 2003-2019 | GRA | boreal_warm |
| IT-Ren | 46.59 | 11.43 | 1999-2020 | ENF | boreal_cool |
| IT-Tor | 45.84 | 7.58 | 2008-2019 | GRA | boreal_cool |

\* CRO: cropland, GRA: grassland, ENF: evergreen needleleaf forest, DBF: deciduous broadleaf forest

\*\* temp_warm: warm temperate climate, temp_cool: cool temperate climate, boreal_warm: cool boreal climate, boreal_cool:

cool boreal climate

**Appendix B - Sensitivity analysis and optimized parameters in twin DA experiments**

The Morris method determines incremental ratios (i.e. elementary effects: EE), from which the mean (μ) and standard deviation (σ) for all the trajectories are calculated to assess the overall importance of each parameter. The μ values are used to rank the parameters in order to systematically identify the non-sensitive parameters (low μ values) from the sensitive ones (high μ). The σ values are used to examine the non-linear effects and/or interactions with other parameters. To assess the results, we look at the normalized μ divided by the maximum μ corresponding to the most sensitive parameter. So thus, a ranking is built with values between 0 and 1, with 1 representing the most sensitive parameters and 0 parameters with no sensitivity (as shown in the color bar of Figure B1). Similarly, the σ values are normalized by the maximum μ. We select the parameters with an average (across LST constraints) normalized μ or normalized σ higher than 0.2. Figure B1 illustrates the 19 LST-related parameters of ORCHIDEE as a result of preliminary sensitivity analyses in the ES-Abr site for 2018, from which we identified 11 parameters (in bold in Table 1 in Sect. 2.3.2) to be optimized in the twin experiments.

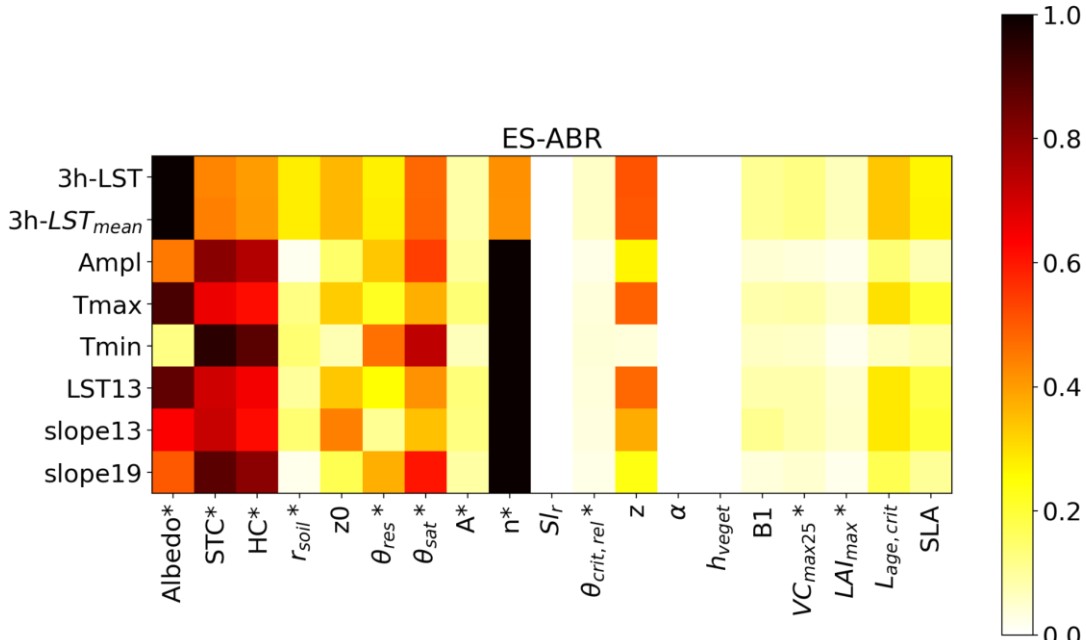

**Figure B1. Morris sensitivity scores obtained in the different SA experiments performed for the ES-Abr site. The plot highlights the most influential key parameters, the various LST features studied and their normalized importance. The LST features studied are: the 3-hourly LST (3h-LST) series (3h-LST), daily mean LST (3h-LST$_{mean}$), daily amplitude LST (Ampl), maximum (Tmax) and minimum (Tmin) LST, LST at 13 local time (LST13), morning (slope13) and afternoon (slope19) gradients.**

**Appendix C - Twin experiments using actual CCI-LST availability and uncertainties**

This approach allows us to conduct twin DA experiments resembling a real observation case, building upon the findings of the previous evaluation that utilized the full availability of pseudo-observations. For that purpose, the 3-hourly LST pseudo-observations, once filtered and perturbed with CCI-LST availability and uncertainties, is used to conduct two DA experiments with the GA method: assimilating the 3-hourly LST series alone (LST DA) and incorporating the Tmax (LST+Tmax DA). These two scenarios were kept since they showed the best results in the previous twin DA experiments. For both DA experiments, the RMSD values are comparable to those obtained when considering the full pseudo-data series, although larger, particularly for LE (Fig. C1 in Appendix). The highest performances across all three variables (LST, LE, and H) are observed in the 3h-LST+Tmax DA experiment, with mean RMSD reductions of 65%, 50%, and 83% in LST, LE, and H, respectively. It is noteworthy that the availability and noise introduced from CCI data have a more significant impact on LE compared to LST and H. Furthermore, assimilating 3-hourly LST alone occasionally results in some runs that increase errors for LE after the optimization. In terms of the optimized parameters, they agree with the optimization using the full pseudo-observation series, i.e., the most sensitive parameters align with the 'true' parameter values with exception of Albedo* (Fig. C2 in Appendix). Despite this, the availability and uncertainties of CCI-LST data may impact significantly the estimation of daily maximum LST (Tmax) − as well as the other characteristics − especially in sites characterized by climates with fewer occurrences of cloud-free conditions, unlike the Mediterranean site used in the twin experiment. Consequently, we will conduct the actual data DA experiments by only considering the entire 3-hourly CCI-LST series (LST DA).

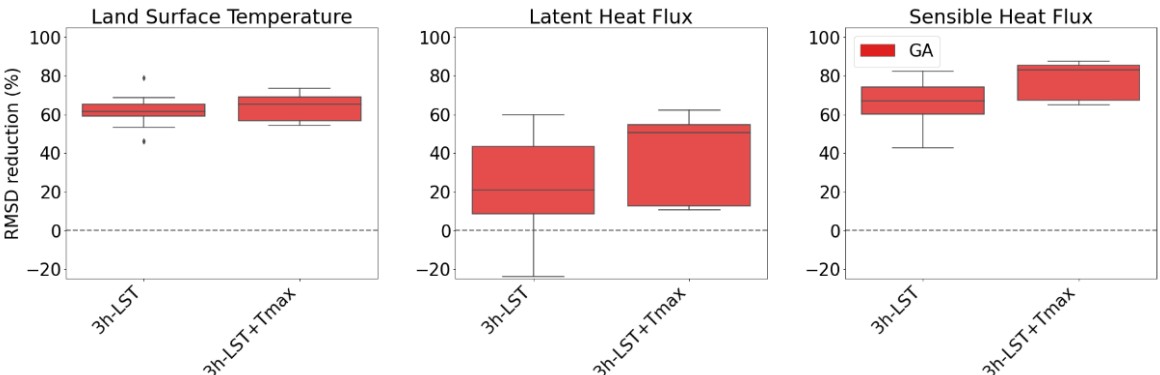

**Figure C1. Boxplots of the model–data RMSD reduction (%) for half-hourly LST, LE and H obtained within 16 optimization tests with random first-guess parameter values using the GA method in the twin experiments considering actual availability and uncertainties from CCI-LST data. The x-axis indicates the experiment assimilating 3-hourly LST pseudo-observations: 3h-LST only and 3h-LST+Tmax.**

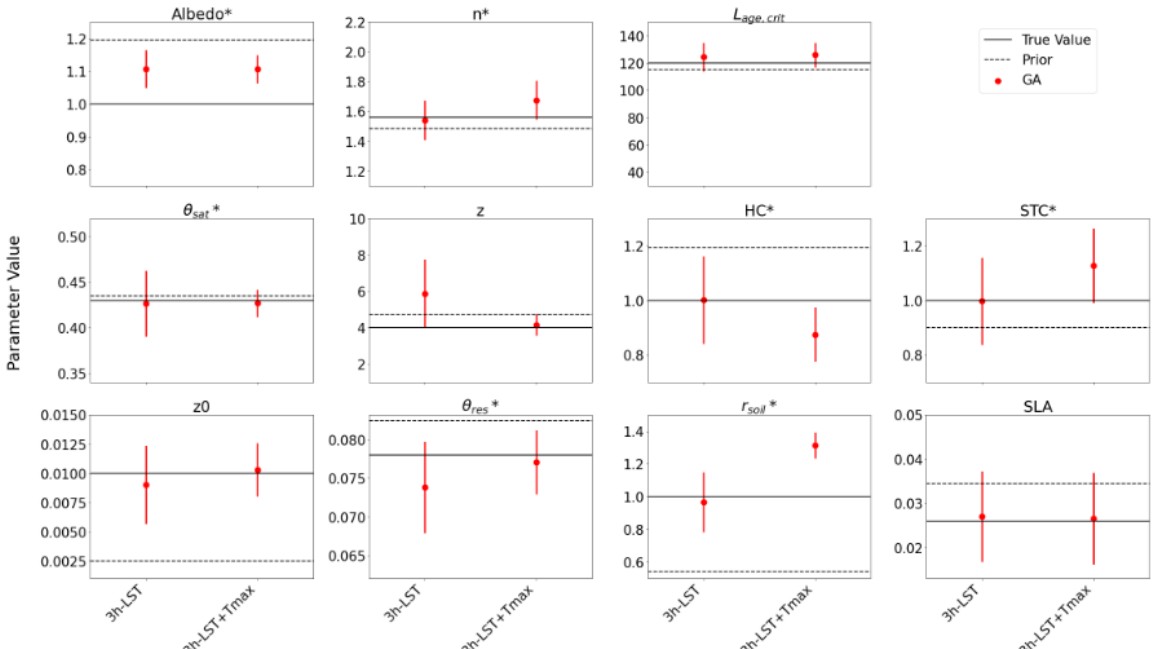

**Figure C2. Parameter estimates for each twin experiment using the GA method considering actual availability and uncertainties of CCI data represented by the mean and standard deviation across 16 optimization tests with random first-guess parameter values. The x-axis indicates the experiment assimilating 3-hourly LST pseudo-observations: 3h-LST only and 3h-LST+Tmax. The "true" parameter (default ORCHIDEE value) and prior values (defined randomly) are represented by the thick and dashed-thin horizontal lines, respectively.**

**Appendix D - Decomposition of mean square errors per site**

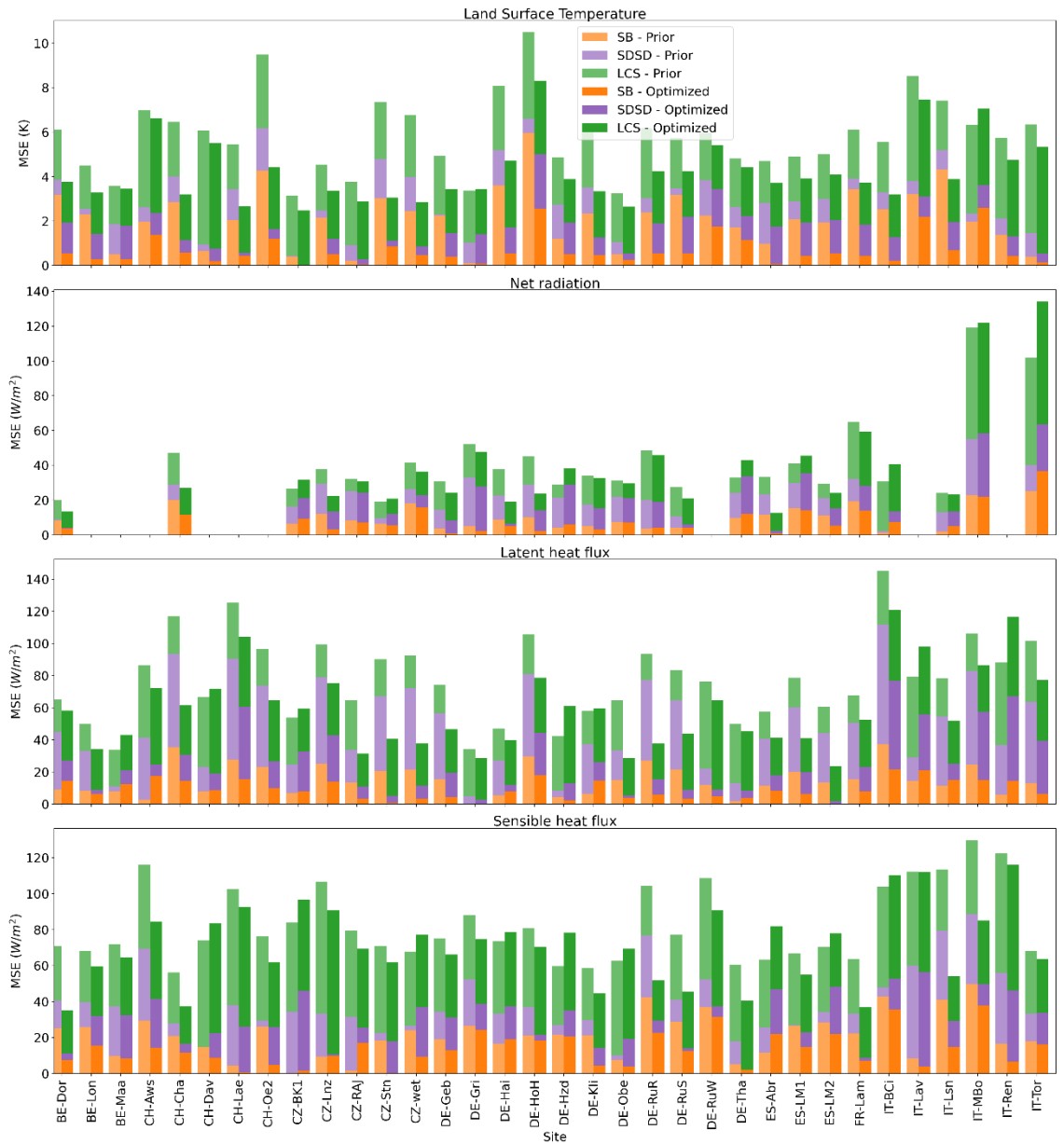

**Figure D1. Decomposition of MSE in terms of bias (SB), lack of correlation (LCS) and difference of standard deviations (SDSD) between the model and observations over sites for LST, Rn, LE and H. The light and dark bars represent the decomposition for prior and optimized simulations, respectively. LST observations are from 3-hourly CCI-LST data, while energy fluxes are from half-hourly in situ observations.**

**Appendix E – Impact of the optimization period**

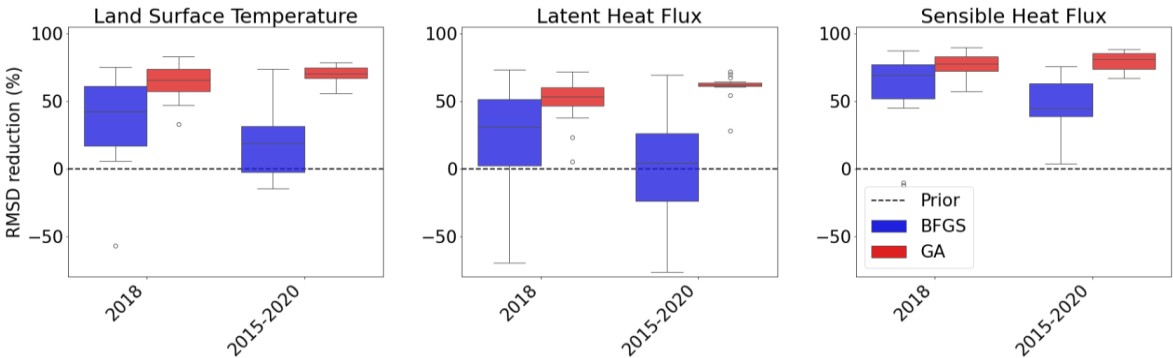

**Figure E1. Comparison of model performance in 2017 for 30-min LST, LE and H when parameters are calibrated in 2018 only or for the entire period (2015-20220) over the selected site in Spain (ES-Abr). Boxplots obtained within 16 optimization tests with random first-guess parameter values for the DA experiment using the gradient-based (in blue) and genetic (in red) methods in terms of model–data RMSD. The DA experiment assimilates the daily mean, amplitude and maximum LST.**

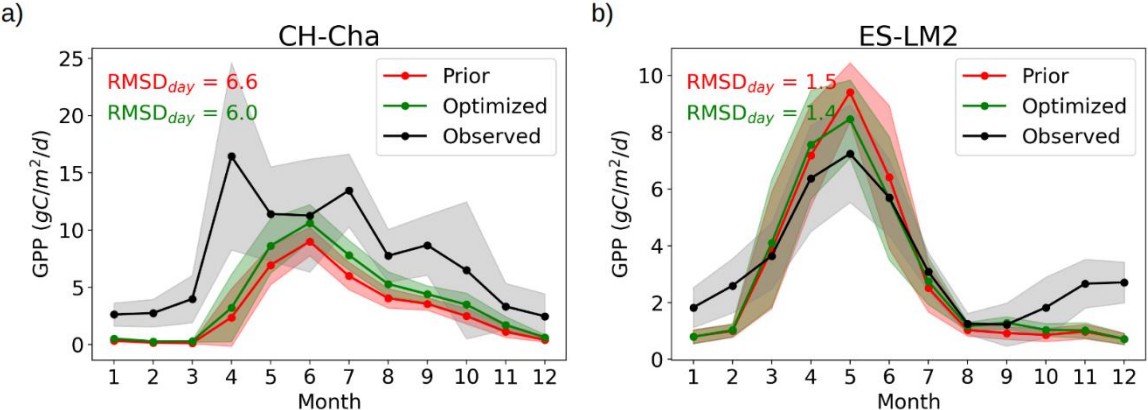

**Figure F1. Annual cycle of GPP modeled for 2018 over a grassland (a: CH-Cha) and cropland (b: ES-LM2) site. The mean (dot) and standard deviation (shaded area) are represented for in situ observations (black), Prior (red) and Optimized (green) ORCHIDEE simulations. The RMSD on the daily basis (RMSD$_{day}$) against in situ observations is shown for Prior (red) and Optimized simulations.**

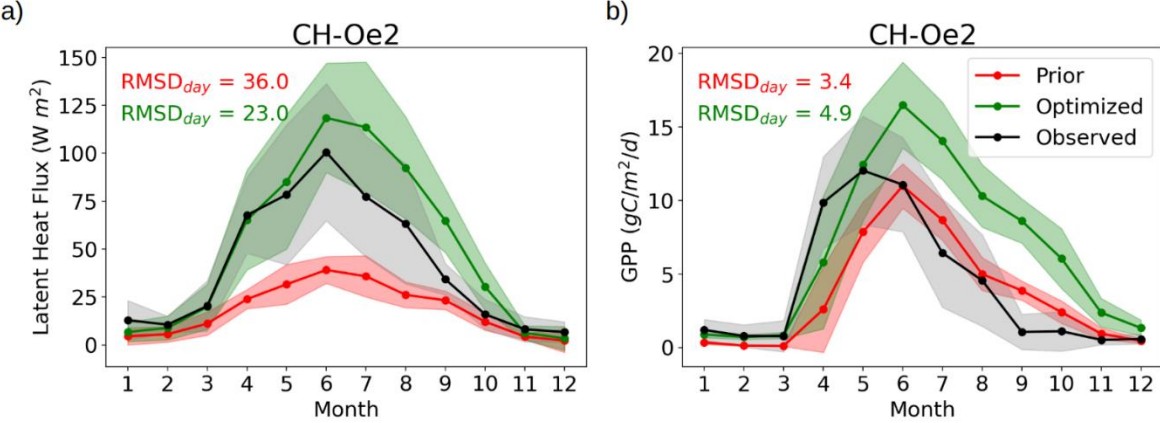

**Figure F2. Annual cycle of LE (a) and GPP (b) modeled for 2018 over a cropland site (CH-Oe2) where LE is improved while GPP is degraded after assimilating CCI-LST data. The mean (dot) and standard deviation (shaded area) are represented for in situ observations (black), Prior (red) and Optimized (green) ORCHIDEE simulations. The RMSD on the daily basis (RMSD$_{day}$) against in situ observations is shown for Prior (red) and Optimized simulations.**