# Peer review of "Assimilating ESA-CCI Land Surface Temperature into the ORCHIDEE Land Surface Model: Insights from a multi-site study across Europe"

_EGUsphere, 2024_

## Author Comment (AC1)

RC1: ['Comment on egusphere-2024-546'](), Anonymous Referee #1, 19 Apr 2024

The manuscript by Olivia-Guerra et al. investigates the impact of assimilating satellite-retrieved land surface temperature (LST) data into a land surface model, here 'ORCHIDEE'. The authors investigate two different optimization methods for the assimilation, a gradient descent method and a genetic algorithm. The employed methods and experiments are well presented and the results are conclusive: assimilating 3-hourly CCI-LST data into ORCHIDEE and employing GA optimization, which proved to be the consistently superior method, lead to a median reduction of 20% in LST and turbulent fluxes (data from 34 sites and for a 12 year validation period). The manuscript is well structured and written and should be published after the minor points listed below have been addressed.

**Reply on RC1**

**We appreciate the time and effort put into reviewing the paper. We addressed all the specific points in the manuscript. They are listed below, followed by our responses in bold and the quotations we propose to add to the manuscript in italics.**

Specific points:

line 102: 'The CCI-LST observations  have an associated ...'

**R: Done.**

line 143: '..., each with increasing grid spacing ...'

**R: Done.**

lines 152-154: consider reformulating these two sentences - they are difficult to understand.

**R: The two sentences were reformulated as follows:**

*"A prior spin-up simulation was performed at each site to bring soil carbon pools, vegetation state and soil moisture content to equilibrium. This procedure is applied*

*by running ORCHIDEE for several hundred years recycling the available forcing data with present-day $CO_2$ concentration.*"

line 180: ' ... assimilations initiated with ...'

**R: Corrected.**

line 213: 'Based on this test, ... to represent the parameters.'

**R: Done. Line 213 was corrected as:**

*"Based on this test, we use 20 trajectories and 10 levels to represent the parameter space well."*

line 214: '.... to each site allows to identify sensitive ...'

**R: Done. Line 214 was corrected as:**

*"The Morris method applied to each site allows to identify the most sensitive parameters for each site."*

line 217: 'For the 34 sites, ...'

**R: Done.**

line 219: '... in the sensitivity analysis: model parameter name, description and default values ...'

**R: Done. The caption of Table 1 was corrected as:**

*"ORCHIDEE parameters evaluated in the sensitivity analysis: model parameter name, description and default model parameter values are shown. The 11 sensitive parameters to be optimized over the selected site for the twin DA experiment are indicated in bold."*

line 220: 'The 11 sensitive parameters to be optimized for the ES-Abr site in the twin DA ...'

**R: Done.**

line 235-236: A set of ... to be optimized is then used as prior data in the optimization.

**R: Corrected as:**

***A set of random values for the 11 parameters to be optimized is then used as the prior parameters in the optimizations.***

Equation 2: in my opinion there is a bracket around the terms in front of the '100' missing: currently the equation does not provide values in percent.

**R: That is correct, the equation 2 was corrected as:**

$$RMSD_{reduction} = \left(1 - \frac{RMSD_{post}}{RMSD_{prior}}\right)100$$

Caption Table 2: 'Example of typical ORCHIDEE parameters optimized in the DA experiments and used to determine the optimum strategy.

**R: The caption was corrected as proposed.**

line 279: '... into which specific error components are improving ...'

**R: Corrected as proposed:**

***"Additionally, we employ the decomposition of the Mean Square Error (MSE: Kobayashi and Salam, 2000) to gain deeper insights into which specific error components are improving or degrading."***

Figures 8 and 9: in my opinion the y-axes of the sub-plots should be labelled 'RMSE' instead of 'MSE'; similarly, 'MSD' on the x-axes should be 'RMSD'. In the caption it should then also say 'root mean square difference' (RMSD), for which the definition could be repeated here for clarity.

**R: The label 'MSE' was changed to 'RMSE'. The following phrase was added to the caption of Figures 8 and 9:**

***"For clarity, note that the square root of error components is plotted."***

---

## Author Response (AR1)

Editor decision: Publish subject to revisions (further review by editor and referees)

Your manuscript "Assimilating ESA-CCI Land Surface Temperature into the ORCHIDEE Land Surface Model: Insights from a multi-site study across Europe" has been subjected now to review by two reviewers. The two reviewers recommend minor revisions. The paper is of good quality and presents an interesting analysis. However, some improvements should be made. This concerns the discussion of the results, which should be in an independent section and take a broader perspective. He authors should also evaluate whether they can quantify the impact of assimilation on the characterization of vegetation processes. I recommend a minor to moderate revision of the manuscript, before it can be published in HESS.

**Reply on Editor decision**

**First of all, we would like to thank the Editor and Reviewers for their valuable suggestions and insightful comments. We have done our best to clarify all the issues raised by the Editor and Reviewers. We included a new version of the manuscript and the point-by-point responses (highlighted in bold) to all comments. Additionally, we have revised the manuscript to improve the overall grammar and readability.**

**In this new version of the manuscript, we included an independent Discussion section taking a broader perspective, where we addressed different points raised by the reviewers, such as: i) the impact of the selection of the optimization period, ii) the assessment of the impact of the assimilation on vegetation processes and soil water availability, and iii) perspectives about assimilation on a regional scale and joint assimilation. The discussion section together with the related appendices of the new version of the manuscript are detailed below.**

**"*4 Discussion***

[revised manuscript text omitted]

In the conclusion of the new version of the manuscript, we also added a paragraph considering the discussion added as follows:

"However, assimilating LST alone has limitations and cannot improve all variables controlling water, energy, and carbon fluxes. Nevertheless, our findings reveal promising outcomes, such as the clear improvement in soil moisture in the twin experiment and the enhancement of GPP in 41% of the studied sites. Despite the challenges, these results indicate that LST data can positively influence variables less directly linked to energy fluxes. This underscores the potential of combining LST with other satellite-derived products, such as leaf area index, albedo, and soil moisture, to better constrain and improve the overall performance of the ORCHIDEE model."

The corresponding appendices for the discussion in the new version of the manuscript were added as follows:

***Appendix E – Impact of the optimization period***

[Figure]

**Figure E1. Comparison of model performance in 2017 for 30-min LST, LE and H when parameters are calibrated in 2018 only or for the entire period (2015-20220) over the selected site in Spain (ES-Abr). Boxplots obtained within 16 optimization tests with random first-guess parameter values for the DA experiment using the gradient-based (in blue) and genetic (in red) methods in terms of model–data RMSD. The DA experiment assimilates the daily mean, amplitude and maximum LST.**

**Appendix F – Impact of assimilating LST on phenology**

[Figure]

**Figure F1. Annual cycle of GPP modeled for 2018 over a grassland (a: CH-Cha) and cropland (b: ES-LM2) site. The mean (dot) and standard deviation (shaded area) are represented for in situ observations (black), Prior (red) and Optimized (green) ORCHIDEE simulations. The RMSD on the daily basis (RMSD$_{day}$) against in situ observations is shown for Prior (red) and Optimized simulations.**

---

## Author Response (AR2)

Editor decision: Publish subject to minor revisions (review by editor)

**We thank the Editor for their suggestions and comments. We addressed the raised points followed by our responses in bold and the quotations we propose to add to the manuscript in italics.**

- I found that the fact that GPP improved at 41% of the sites is over-emphasized in the paper. For the majority of the sites a deterioration was observed. Could you modify the discussion to take this into account?

**R. This is considered in the Discussion and Conclusion sections. In the Discussion, the overall degradation in GPP is more clearly stated and further illustrated with an additional figure in the Appendix, which highlights the degradation in GPP alongside observed improvements in energy fluxes. In the new version of the article, this is modified as:**

*"Regarding the GPP, we assess the impact by assimilating the 3-hourly CCI-LST time series over the 34 WarmWinter sites in 2018. Assimilating the 3-hourly LST data results in an overall degradation in GPP, with a median RMSD increase across sites of 7.4%. Among the studied sites, 14 out of 34 shows improvements in diverse conditions such as the grassland CH-Cha and Mediterranean ES-LS2 sites (see Fig. F1 in Appendix F). At the other sites, larger errors were obtained, with RMSD increases of up to 55.4%. For instance, while the cropland CH-Oe2 site exhibits a 36.1% improvement in LE, the RMSD in GPP is increased by 41.3% (see Fig. F2 in Appendix F). Despite the mixed results obtained on GPP, the improvement observed in the 14 sites (i.e., 41% of the sites) is a promising outcome, especially considering the challenge of enhancing model variables that are not closely linked to LST."*

[Figure]

*Figure F2. Annual cycle of LE (a) and GPP (b) modeled for 2018 over a cropland site (CH-Oe2) where LE is improved while GPP is degraded after assimilating CCI-LST data. The mean (dot) and standard deviation (shaded area) are represented for in situ observations (black), Prior (red) and Optimized (green) ORCHIDEE simulations. The RMSD on the daily basis (RMSD_day) against in situ observations is shown for Prior (red) and Optimized simulations.*

**In the new version of the conclusions, the improvement in GPP of 41% of the sites is removed:**

*"Nevertheless, our findings reveal promising outcomes, such as the clear improvement in soil moisture in the twin experiment and the enhancement of GPP at several studied sites."*

- A side remark is that it would also be interesting to see how WUE is affected by LST assimilation. But I understand that this might be beyond the scope of this paper.

**R. We recognize that the impact of assimilating LST on WUE is an interesting and closely related topic, however, we agree that in this paper focused mainly on energy surface fluxes, this falls outside the scope of this study. Given the mixed results obtained on GPP, and the need to recalibrate other variables linked to the carbon cycle via the joint assimilation of "biomass" remote sensing products such as LAI or SIF data, we don't expect relevant outcomes in the analysis of the simulated WUE. We agree that this is a very important variable to look at and that it will be done in the following work.**